# A common mechanism of proteasome impairment by neurodegenerative disease-associated oligomers

Tiffany A. Thibaudeau[1], Raymond T. Anderson[1] & David M. Smith [1]

Protein accumulation and aggregation with a concomitant loss of proteostasis often contribute to neurodegenerative diseases, and the ubiquitin–proteasome system plays a major role in protein degradation and proteostasis. Here, we show that three different proteins from Alzheimer's, Parkinson's, and Huntington's disease that misfold and oligomerize into a shared three-dimensional structure potently impair the proteasome. This study indicates that the shared conformation allows these oligomers to bind and inhibit the proteasome with low nanomolar affinity, impairing ubiquitin-dependent and ubiquitin-independent proteasome function in brain lysates. Detailed mechanistic analysis demonstrates that these oligomers inhibit the 20S proteasome through allosteric impairment of the substrate gate in the 20S core particle, preventing the 19S regulatory particle from injecting substrates into the degradation chamber. These results provide a novel molecular model for oligomer-driven impairment of proteasome function that is relevant to a variety of neurodegenerative diseases, irrespective of the specific misfolded protein that is involved.

---

[1] West Virginia University, School of Medicine, Department of Biochemistry, PO Box 9142, Morgantown, WV 26501, USA. Correspondence and requests for materials should be addressed to D.M.S. (email: dmsmith@hsc.wvu.edu)

The most common neurodegenerative diseases are characterized by an accumulation of aggregation-prone proteins concomitant with a loss of proteostasis, which results in progressive death of neurons[1–3]. Culminating evidence from the past two decades has revealed that soluble, oligomeric forms of protein aggregates (such as Aβ in Alzheimer's disease, α-Synuclein (α-Syn) in Parkinson's disease, and mutant huntingtin in Huntington's disease) are likely the most toxic species[4, 5]. While different regions of the brain are affected in these distinct diseases, proteotoxicity is a shared feature found in these affected regions of the brain. This suggests that a common mechanism of proteotoxicity could contribute to the development and progression of these distinct neurodegenerative diseases.

Proteostasis[6, 7] is maintained by several systems in the cell including the ubiquitin–proteasome system (UPS), chaperones, chaperone-mediated autophagy, and macroautophagy[8]. The UPS is the principal route for the degradation of intracellular misfolded, damaged, or unneeded proteins[9]. If the efficiency of proteostasis systems declines, misfolded proteins accumulate and aggregate in the cell, which can disrupt normal cellular functions and even cause cell death[10]. Maintaining proteostasis is especially important for neurons due to their complex architecture, long lifespan, and inability to dilute aggregate load by cell division[11]. Most importantly, the UPS is critical for normal functioning of neuronal synapses, including synaptic protein turnover, plasticity, and long-term memory formation, which rely on tightly controlled changes in the proteome[11–15]. Recently, Ramachandran and Margolis[16] identified a mammalian nervous-system-specific membrane proteasome complex that directly and rapidly modulates neuronal function by degrading intracellular proteins into extracellular peptides that stimulate neuronal signaling through postsynaptic N-methyl-D-aspartate receptors.

Decreased proteasome function has been reported in a broad array of chronic neurodegenerative diseases[17]. Impaired proteasome function has been implicated, as a primary cause or a secondary consequence, in the pathogenesis of many neurodegenerative diseases, including Alzheimer's, Parkinson's, and Huntington's diseases[2, 17–21]. In fact, brain region-specific proteasome inhibition (e.g., forebrain, substantia nigra) closely mirrors the neuropathology and clinical hallmarks of neurodegenerative diseases[22–26]. A small percentage of neurodegenerative disease is caused by hereditary gene mutations, many of which affect components of the UPS (e.g., PARK1, PINK)[20]. However, the vast majority of neurodegeneration is idiopathic in origin and the involvement of the UPS is less clear[17]. What is clear in these diseases is that proteins that are normally degraded are not properly degraded after misfolding occurs, leading to their accumulation. Several groups have provided evidence that aggregated proteins from neurodegenerative diseases interact with and impair proteasome function[27–37]. However, it is not clear what specific types of aggregates impair the proteasome, and a mechanistic understanding of how they do so has not been elucidated. Though, one study has been able to show that heterogeneous aggregates of the mouse prion protein, PrP^sc, reduced substrate entry by decreasing proteasomal gating[36]. Despite these many efforts, an understanding of why and how the proteasome is so generally impaired in neurodegenerative disease has remained elusive. Understanding the mechanism of impairment will provide a basis for drug development to restore proteasome activity and proteostasis in the brain and is therefore an important effort.

Proteins targeted for proteasomal degradation are marked by the attachment of several ubiquitin proteins. These polyubiquitinated substrates are recognized by the 26S proteasome and are degraded[9]. The 26S proteasome is made up of a 20S proteasome core particle capped on one or both ends by the 19S regulatory particle. It degrades proteins by a multistep process: the 19S regulatory particle binds ubiquitinated substrates and opens a substrate entry gate in the 20S[38–40] and unfolds its substrates by translocating them into the 20S catalytic chamber were they are degraded[41, 42]. The 20S is a hollow cylindrical complex composed of four heteroheptameric rings arranged in a $\alpha_7$-$\beta_7$-$\beta_7$-$\alpha_7$ fashion[43]. Proteolysis occurs on the interior surface of β-subunit rings. The substrate gate is formed by the N-termini of the α-subunits, which prevent unregulated access to the catalytic sites by folding over the entry pore and blocking substrate translocation into the catalytic chamber[44]. Triggering of gate opening by the 19S requires the C-terminal HbYX motif of the 19S ATPases to bind to intersubunit pockets (between the α-subunits) on top of the 20S[45]. The HbYX motif allows the 19S to bind to the 20S core particle, but binding of the HbYX motif by itself (as a hepta peptide) is also sufficient to allosterically induce conformational changes in the α-subunits that cause gate opening[45–48]. Clearly, regulation of the 20S proteasome gate is an important aspect of proteasome function and the cell has evolved many different proteasomal regulators that control 20S gate opening, many of which contain the HbYX motif (e.g., the 19S ATPases: Rpt2, Rpt3, Rpt5; Blm10/PA200; Pba1–Pba2; PI31; and archaeal CDC48/P97), and some that do not (i.e., the 11S family: PA28αβ and PA26)[48].

This study demonstrates that misfolded proteins from three distinct neurodegenerative diseases adopt a common three-dimensional (3D) conformation that is capable of impairing ubiquitin-dependent and ubiquitin-independent proteasome function. Although these oligomers possess unique primary sequences, they all impair the proteasome through allosteric stabilization of the closed gated conformation of the 20S core particle, therein blocking protein degradation. Moreover, these toxic oligomers specifically impair HbYX motif dependent gate opening, yet do not impair gate opening induced by the 11S family of regulators. These data suggest that proteasome impairment in various neurodegenerative diseases may share a common mechanism.

## Results

**Inhibitory oligomers share structural features.** Prior studies report conflicting observations regarding the impairment of the proteasome by disease-related aggregated proteins, some demonstrating proteasome impairment[27–30, 32–35], while others do not[49, 50]. The major limitation of these studies is that the conformational state of the aggregates was not accounted for or considered. Aggregation-prone proteins have the unique property of conformational polymorphism. During amyloid formation a variety of aggregate species are formed, ranging from small dimers up to large insoluble fibrils. Oligomers are metastable intermediates to fibril formation or an off-pathway product of aggregation and are recognized as the primary pathogenic effectors[51]. Since the previous studies used heterogeneous compositions of the aggregated proteins, these seemingly conflicting results for proteasome impairment are not surprising. In this study, we purify to homogeneity a specific conformation of a pathological oligomer, identified its conformational status, and extensively characterize its mechanism of impairment on human and mammalian 20S/26S proteasomes. In addition, this study identifies a specific oligomeric conformation found in Alzheimer's, Parkinson's, and Huntington's disease that substantially impairs proteasome function in a way that could contribute to the development and progression of these and other neurodegenerative diseases.

To determine if specific types of oligomers are responsible for proteasome impairment, we began by generating various mixed

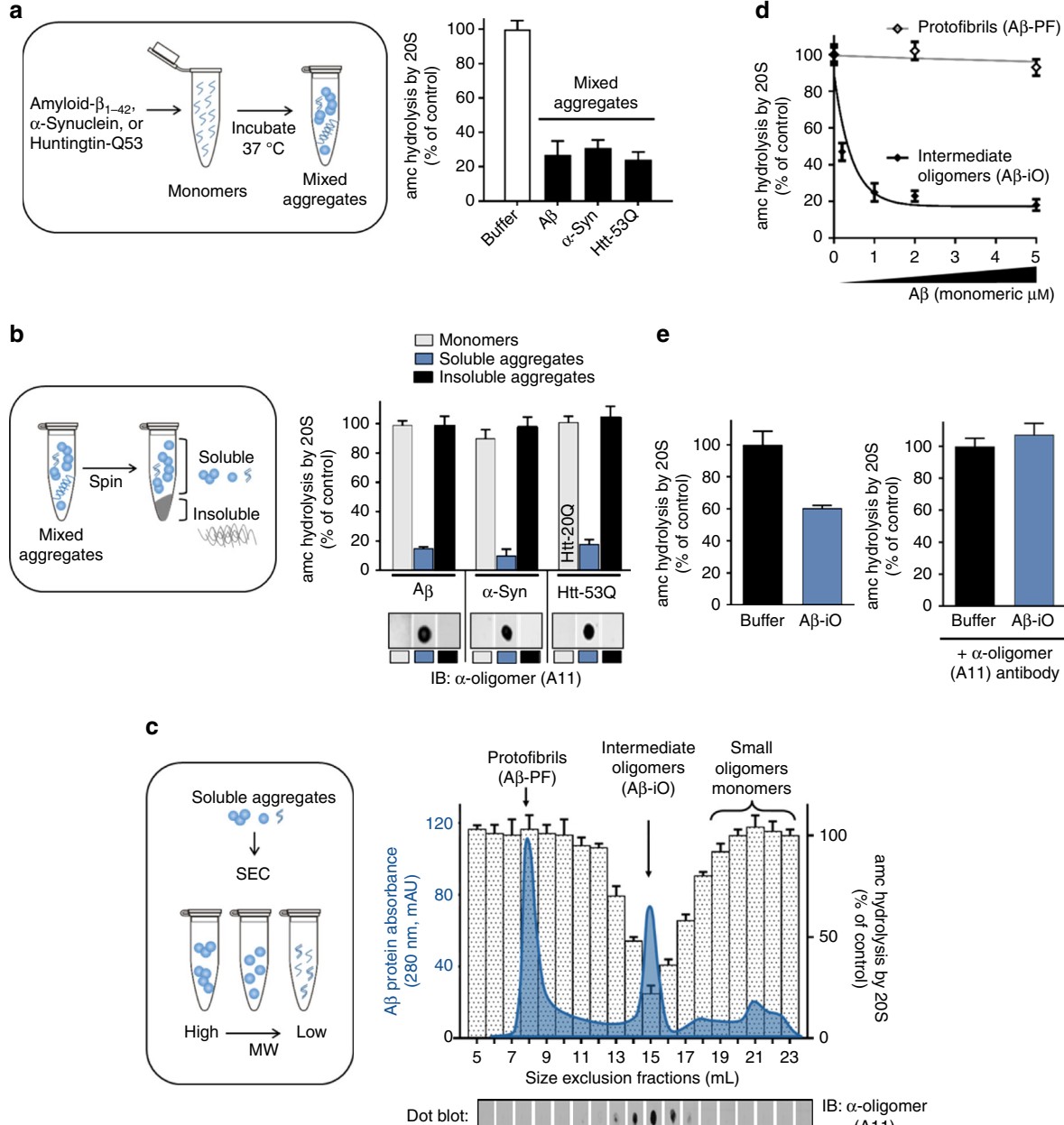

**Fig. 1** A specific conformation of soluble oligomers potently inhibits the mammalian 20S proteasome. **a** Mammalian 20S proteasomes were incubated with mixed aggregates of Aβ$_{1-42}$ (5 μM), α-Syn (1 μM), Htt-53Q (0.1 μM), or an equal volume of oligomer buffer (control). Proteasome activity (linear rate of LLVY-amc hydrolysis) is represented as a percentage of activity compared to the control. **b** Crude aggregates from **a** were separated into soluble and insoluble aggregates (schematic, left) and were assayed as in **a** (bar graph, right). For huntingtin monomers, Htt-20Q monomers were used because pure Htt-53Q monomers could not be obtained due to rapid oligomerization. Dot blots of monomers, soluble aggregates, and insoluble aggregates from **b** were probed with the conformation-dependent anti-oligomer "A11" antibody (bottom right). **c** Soluble Aβ aggregates from **b** were separated by size exclusion chromatography (Abs$_{280 nm}$, solid blue line). Two microliters from each fraction was evaluated for its effect on 20S proteasome chymotrypsin-like activity (bars) and probed for anti-oligomer A11 reactivity (dot blot, bottom). **d** Proteasome activity with up to 5 μM of Aβ oligomers (Aβ-iO) or Aβ protofibrils (Aβ-PF) from **c**. **e** Intermediate oligomers from **d** were pre-incubated with anti-oligomer A11 antibody (Aβ-iO + A11) or an equal volume of antibody buffer (Aβ-iO) for 30 min at 37 °C before to addition to proteasome activity assay. Final concentration of Aβ-iO in the assay was 0.25 μM (the ~IC$_{50}$ as determined in **d**). The concentrations of aggregates are calculated based on the respective monomeric peptide/protein mass (Aβ, 4.5 kDa; α-Syn, 14 kDa; and Htt-53Q, 22 kDa). All controls contained an equal volume of buffer identical to that of the respective aggregates. The data are representative of three or more independent experiments performed in triplicate. Error bars represent ± standard deviation

populations of protein aggregates made from either amyloid-β 1–42 (Aβ), α-Syn, or huntingtin exon 1 with a polyQ-expansion (Htt-53Q) and asked if they could impair purified mammalian 20S proteasome. We found that under specific oligomerization conditions (different for each protein type) each of the aggregate preparations could significantly impair the 20S proteasomes

ability to hydrolyze fluorogenic peptide substrates (Fig. 1a). These results replicate those which have been reported to some extent previously[29–31]. Next, we separated the mixed aggregates into soluble and insoluble fractions and again tested their effect on proteasome activity. The soluble oligomers, but not equal amounts of monomers or insoluble fibrils, strongly impaired

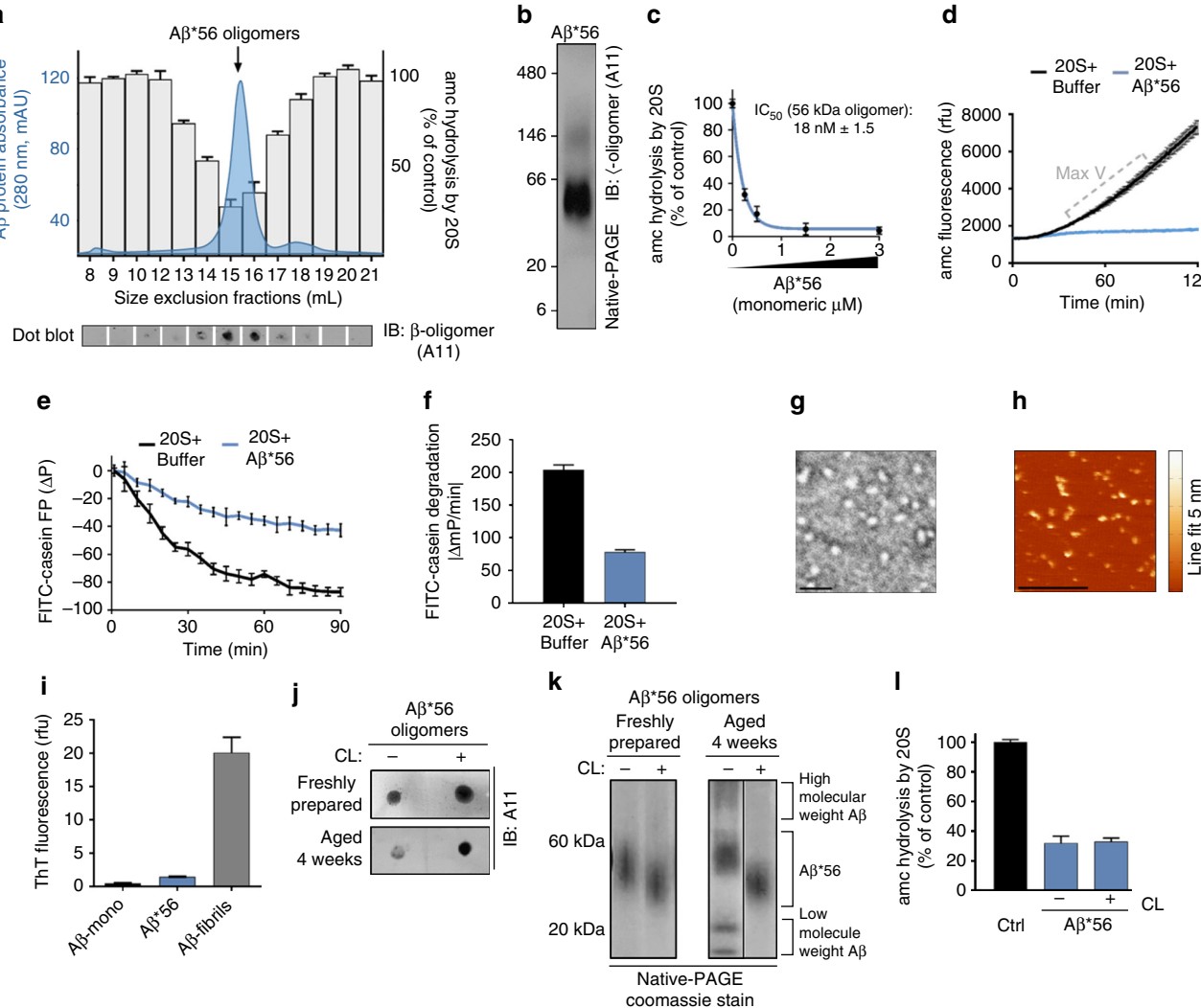

**Fig. 2** Generation of stabilized A11+ oligomers (Ab*56) for mechanistic evaluation of proteasome inhibition. **a** Size exclusion chromatography (Superose 6 GL 10/300) run that was used to generate a pure Aβ*56 oligomer preparation (blue solid line, left axis), with proteasome activity (amc hydrolysis) from 1 μl of the corresponding fractions (bars, right axis; as in Fig. 1), and A11 dot blot (bottom panel). **b** Native-PAGE Aβ*56 peak fraction from **a** followed by western blot using the A11 anti-oligomer antibody. **c** Proteasome activity (LLVY-amc hydrolysis) with titrating Aβ*56; the IC$_{50}$ is ~0.22 μM Aβ (4.5 kDa monomeric mass) or 18 nM Aβ*56 oligomer complexes (56 kDa mass each). **d** Representative raw data of proteasome activity assay (nLPnLD-amc hydrolysis) with 1.5 μM Aβ*56 oligomers (data point from **c**). **e** Change in polarization of FITC-labeled-casein protein (due to cleavage) in the presence of the 20S proteasome, with and without 10 μM Aβ*56. **f** Rate of FITC-casein degradation (ΔmP/min). **g** Representative negative stain electron microscopy image of purified Aβ*56 oligomers. Scale bar is 25 nm. **h** Representative tapping mode atomic force microscopy topography image of Aβ*56 oligomers. Scale bar is 0.5 μm. Heat map for oligomer height is shown on the right. **i** ThT fluorescence of the indicated Ab preparations. **j, k** Dot blot with A11 antibody **i** and oligomer native gel electrophoresis visualized by Coomassie stain **k** of non-crosslinked and glutaraldehyde crosslinked (CL) Aβ*56 oligomers before and after 4-week incubation at 4 °C. **l** Proteasome activity (nLPnLD-amc hydrolysis) in the presence of CL and non-crosslinked Aβ*56 oligomers (0.75 μM). Concentration of Aβ*56 oligomers is calculated based on the mass of peptide monomer (Aβ, 4.5 kDa). All controls contained an equal volume of buffer identical to that of the respective aggregates. Proteasome activity was calculated as in Fig. 1a. The data are representative of three or more independent experiments performed in triplicate. Error bars represent ± standard deviation. All following experiments utilize CL Aβ*56 oligomers unless indicated otherwise

proteasome activity (Fig. 1b) in a concentration-dependent manner (Supplementary Fig. 1). The eukaryotic proteasome has three types of active sites, each displaying preference for cleavage after specific residues (chymotrypsin like, hydrophobic; caspase like, acidic; trypsin like, basic). Substrate hydrolysis by all three of the catalytic sites were impaired by the soluble oligomers (Supplementary Fig. 2).

Aβ, α-Syn, and Htt-53Q monomers are relatively unstructured and they can enter the 20S proteasome to be degraded. However, the monomers fail to impair peptide hydrolysis by the proteasome at equal concentrations as the oligomers (Fig. 1b)

therefore, substrate competition at the active site cannot explain impairment by the oligomers. Furthermore, oligomers are too large (Fig. 2g, h) to enter the 13 Å wide substrate-entry channel of the 20S proteasome. Additionally, since the insoluble aggregates of these proteins cannot impair the proteasome (Fig. 1b), this suggests that impairment by the oligomers may be due to a specific conformation of the soluble oligomers, which is lost after conversion to larger aggregates or fibrils. This is consistent with literature that ascribes cellular toxicity to soluble oligomers in neurodegenerative diseases[4, 52, 53]. Many species of oligomeric structures have been described, and antibodies

developed to recognize specific structural conformations of disease-related species[54–56]. Kayed et al.[55] generated a polyclonal anti-oligomer antibody (A11) that specifically recognizes some types of protein oligomers independent of the proteins amino acid sequence. This A11 antibody recognizes some oligomeric species of Aβ, polyglutamine proteins, α-Syn, and prion, and has been used to assess the presence of oligomers in diseased brains compared to aged matched controls[55, 57]. We performed a dot blot with A11 on the monomers, oligomers, and insoluble fibrils for each protein that we tested. All three of the soluble oligomer preparations contained the A11 epitope (A11+), while the epitope was absent in the monomeric and fibril fractions (Fig. 1b bottom).

It is interesting that all three soluble oligomer types that impaired proteasome activity also showed strong A11 antibody binding. To correlate proteasome impairment with the presence of the A11 epitope more specifically, the soluble fraction of the Aβ aggregates were separated by size exclusion chromatography (SEC). Three prominent populations of soluble aggregates were observed, one in the void volume consistent with larger protofibrils (Aβ protofibrils (Aβ-PF)), a second peak corresponding to intermediate-sized oligomers (Aβ-iO), and a third pool of small oligomers and monomers (Fig. 1c chromatogram). The effect of each fraction on 20S proteasome activity was determined. Only the intermediate-sized oligomers (~56 kDa) impaired the 20S proteasome (Fig. 1c bars) and this impairment correlated with the fractions that were positive for A11 (Fig. 1c dot blot). This inhibitory species also impaired the degradation of fluorogenic substrates specific for each of the 20S's three different proteolytic sites as observed in the mixed oligomer populations (Supplementary Fig. 3A). This suggests that impairment could be due to impairment of substrate entry rather than impairment of a specific catalytic active site. The Aβ protofibril peak (Aβ-PF), lacking the A11 epitope, did not impair degradation of any fluorogenic substrates, even in the presence of ten times more Aβ-PF than Aβ intermediate oligomers (Fig. 1d and Supplementary Fig. 3B).

It is plausible that proteasome impairment is due to the oligomer size rather than a specific oligomeric structure. To determine if the impairment is due to the size of the oligomer/protofibrils and whether the shared A11 reactivity is merely a coincidence, we generated high-molecular weight (200–400 kDa) A11+ Aβ oligomers (Aβ-iO) (Supplementary Fig. 4A) and A11+ Aβ-PF (>700 kDa) (Supplementary Fig. 5A). The high MW A11+ Aβ-iO impaired the 20S commensurate with the level of A11 reactivity (Supplementary Fig. 4A & B). The higher molecular weight A11+ Aβ-PF also impaired substrate hydrolysis by all three active sites of the proteasome (Supplementary Fig. 5B), although to a considerably lesser extent than the intermediate Aβ-iO (Fig. 1d). This is expected based on the proposal that protofibrils form when the oligomers bind to one another to form a chain of oligomers[58], which sterically blocks surfaces on the internal oligomers but not the terminal ones, which could still interact with the proteasome. To further determine if the structural epitope of the A11 antibody on the intermediate Aβ-iO is necessary for proteasome impairment we performed a neutralization assay to block the A11 epitope. A11+ oligomers were incubated with the A11 antibody prior to testing proteasome activity. Aβ-iO were used at a concentration of 0.2 μM, the IC50, as determined in Fig. 1d, so an increase or decrease in proteasome activity could be readily observed. Indeed, the A11 antibody when bound to the Aβ-iO completely rescued proteasome activity (Fig. 1e). As a control, the experiment was repeated with an antibody raised against the N-terminal residues of Aβ (clone NAB228), which did not rescue proteasome function (Supplementary Fig. 6). This demonstrates that an available oligomer-specific A11 epitope site is necessary for impairment of the proteasome.

**Characterization of homogenous and stable oligomers.** Above we described the isolation of a specific proteasomal inhibitory oligomer from a mixed population of oligomers and aggregates. In order to determine the mechanism of impairment we sought to generate homogenous, stable, and reproducible A11+ oligomers, which could be used for reliable mechanistic analysis. In contrast to α-Syn and huntingtin aggregates, methods to generate physiological relevant oligomers from synthetic Aβ peptides have been extensively developed. Barghorn et al.[59] characterized a highly stable Aβ (1–42) oligomer species (~dodecamer) which can be prepared in vitro and can be found in the brains of patients with AD. The relevance of dodecameric Aβ-iO to disease pathology is established[57, 60]. With some modifications to the protocol of Barghorn et al.,[59] we generated Aβ*56 oligomers, and purified them by nondenaturing SEC (Fig. 2a chromatogram). The major peak corresponds to the intermediate-sized Aβ-iO in Fig. 1d. We tested each fraction for proteasome activity and found the major peak impaired the 20S (Fig. 2a bar graph). The single symmetric protein peak demonstrates the homogenous nature of the oligomer preparations. Consistent with Aβ*56 oligomers isolated from human brain tissue and cerebrospinal fluid, our Aβ*56 oligomers are A11+ (Fig. 2a), run at ~56 kDa (Fig. 2b)[57], and significantly impair proteasome activity in a concentration-dependent manner (Fig. 2c). Representative real-time fluorogenic substrate hydrolysis data are also shown (Fig. 2d).

Since Aβ*56 can impair 20S peptide substrate degradation, we asked if it could impair protein degradation as well. The 20S core particle by itself cannot unfold proteins, so we used β-casein, a classical unfolded protein substrate. We used FITC-labeled casein to follow its degradation in real time using anisotropy, which monitors the tumbling rate of the fluorophore. When FITC-labeled casein is degraded by the proteasome, the tumbling rate of the fluorophore increases, causing a decrease in anisotropy (Fig. 2e 20S + buffer). Similar to peptide substrates, purified Aβ*56 oligomers also impaired proteasome degradation of the FITC-labeled casein protein (Fig. 2e 20S + Aβ*56), demonstrating that Aβ*56 also impairs the degradation of an unfolded protein. We confirmed that the relevant morphology of these oligomers were consistent with those published for synthetic and human brain derived oligomers via: native gel electrophoresis[57, 61] (for MW), transmission electron microscope (TEM)[62] (for spherical shape), atomic force microscopy (AFM)[63, 64] (for size), Thioflavin-T (ThT) staining[55] (slight but low staining), and anti-oligomer immuno-detection[57] (Fig. 2g–i).

Oligomers are metastable intermediate structures, which complicates analysis when consistent homogeneous preparations are need for in-depth biochemical analysis. To circumvent this issue, we stabilized the Aβ*56 oligomers by crosslinking, which maintained the conformation of the A11+ epitope for 4 weeks when stored at 4 °C (Fig. 2j) and without crosslinking the A11+ epitope was not as stable over this time period. In addition, the apparent mass of the crosslinked (CL) oligomers was also assessed via Native-PAGE and we found that it was unchanged over the 4-week incubation (Fig. 2k). The CL Aβ*56 oligomers ran slightly faster than the non-crosslinked oligomers as expected[59], likely due to stabilization of the CL structure. In contrast, the non-crosslinked oligomers partially dissociated into smaller oligomers and formed larger oligomers after 4 weeks (Fig. 2k). Most importantly, crosslinking of the oligomers does not alter their proteasome impairment activity compared to the non-crosslinked form (Fig. 2l). Together, this demonstrates that the synthetic Aβ*56 oligomers are homogenous, relevant, stable,

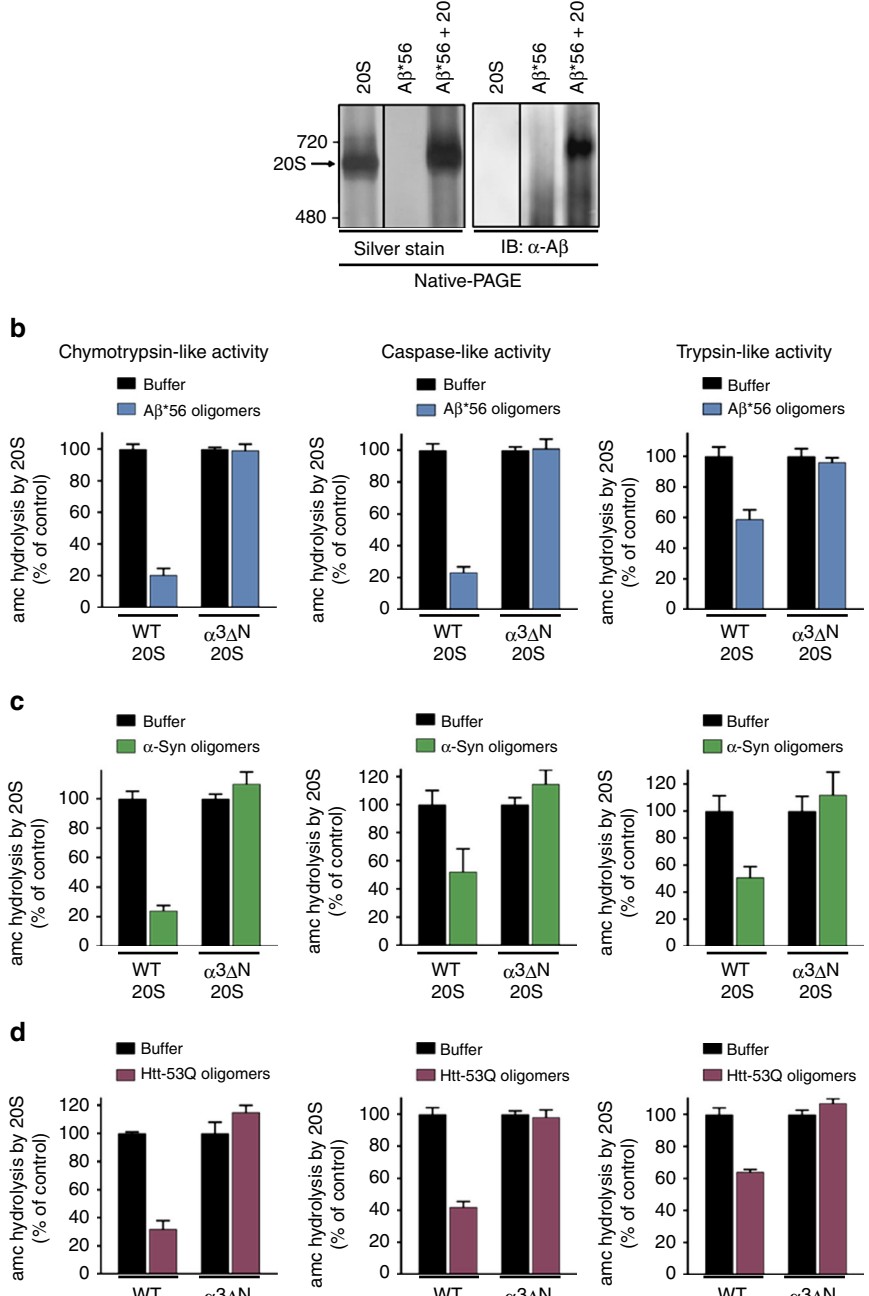

**Fig. 3** The A11(+) oligomers bind to the 20S proteasome and impair opening of the substrate gate. **a** 20S proteasomes (0.4 μg) and pure non-crosslinked Aβ*56 oligomers (1.5 μg) were incubated separately or together for 30 min (37 °C), crosslinked with 1 mM glutaraldehyde for 5 min, and separated by Native-PAGE (4–8% Tris–acetate gel). Total protein was detected by silver stain (left), and total Aβ was detected by western blot (right). **b–d** The activity of yeast 20S wild-type (WT) and open-gate (α3ΔN) proteasomes was measured for all three proteolytic sites in the presence of A11(+) oligomers from Aβ*56 (**b**; 2.5 μM), α-Syn (**c**; 100 nM), and Htt-53Q (**d**; 50 nM). Chymotrypsin-like activity was measured by LLVY-amc hydrolysis, trypsin-like activity by RLR-amc, and caspase like by nLPnLD-amc hydrolysis. The concentrations of aggregates are calculated based on the respective monomeric peptide/protein mass. All controls contained an equal volume of buffer identical to that of the respective aggregates. The data are representative of three or more independent experiments performed in triplicate. Error bars represent ± standard deviation

reproducible, and represent a single oligomeric species that potently impairs peptide and protein degradation by the 20S proteasome. These CL Aβ*56 oligomers are therefore ideally suited for further mechanistic and biochemical analysis to understand how oligomers impair proteasome function and are thus used in all of the following experiments using Aβ-iO unless stated otherwise.

**Direct binding of Aβ*56 to 20S proteasome**. We next sought to determine if Aβ*56 and the 20S proteasome could be observed to directly interact. Non-crosslinked Aβ*56 oligomers were mixed with purified 20S proteasomes. To stabilize their interaction, we used a low concentration of glutaraldehyde (1 mM) to induce crosslinking and analyzed migration. Aβ*56 is clearly seen co-migrating with the 20S proteasome by Native-PAGE gel

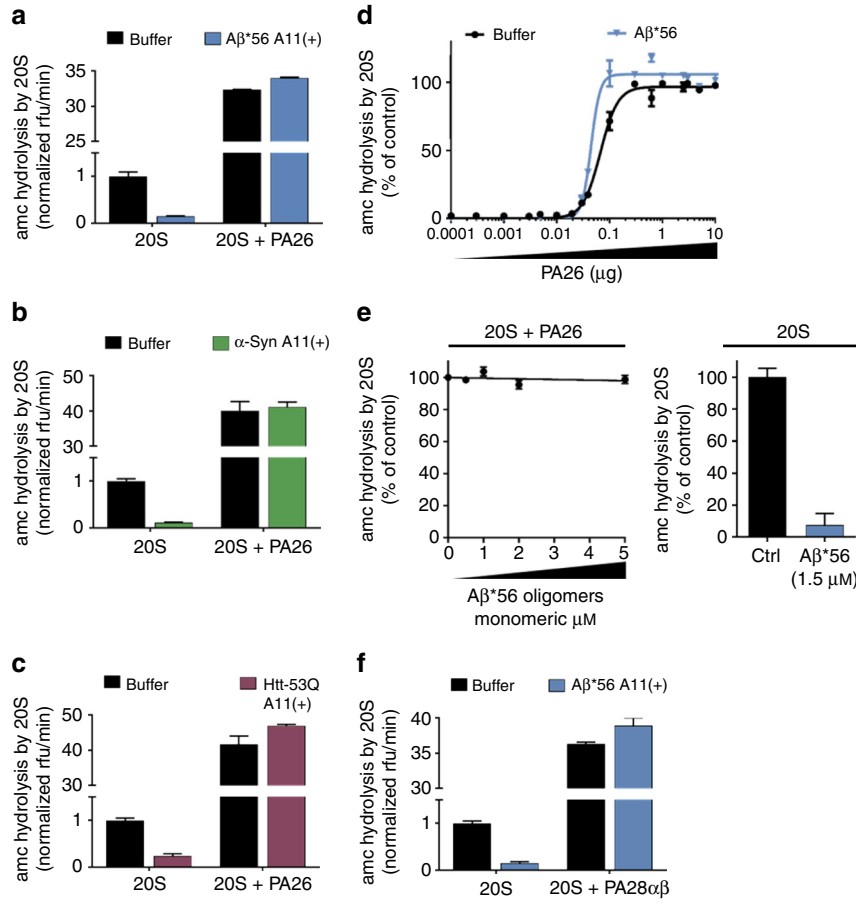

**Fig. 4** A11(+) oligomers cannot inhibit PA26 or PA28αβ induced gate opening. **a–c** 20S Proteasome activity with and without the proteasome activator PA26 (1 μg/100 μl) was determined in the presence of A11(+) oligomers from Aβ*56 (1.5 μM, **a**), α-Syn (0.1 μM, **b**), or Htt-53Q (0.1 μM, **c**). Broken graphs are used to show the extent of 20S inhibition, while still showing the extent of PA26 activation. **d** Proteasome activity in the presence of increasing concentrations of PA26 with and without A11(+) Aβ*56 (1.5 μM). The sigmoidal equation was fit to the averages from three independent experiments (normalized to % activity) performed in triplicate, error bars ± SEM. For **a–d**, the 20S proteasome activity (nLPnLD-amc hydrolysis, rfu/min) was normalized to 20S control activity without activator. **e** Proteasome activity with PA26 activator (1 μg/100 μl) in the presence of increasing concentrations of Aβ*56 (left). Activity of 20S proteasome (without PA26) with 1.5 μM Aβ*56 is shown at right. **f** Same experiment as in **a**, with human PA28αβ replacing PA26. The concentrations of oligomers are calculated based on the respective monomeric peptide/protein mass. All controls contained an equal volume of buffer identical to that of the respective aggregates. The data are representative of three or more independent experiments performed in triplicate. Error bars represent ± standard deviation

visualized with both silver stain (Fig. 3a left) and by immunoblotting for total Aβ (Fig. 3a right). Notably, the low concentration of glutaraldehyde treatment did not cause random nonspecific protein crosslinking and did not crosslink the entire multisubunit proteasome into a single 700 kDa complex as determined by the absence of protein aggregates in the SDS-PAGE stacking gel (Supplementary Fig. 7). Minimal crosslinking conditions are further demonstrated by the discrete banding pattern of multiple subunits and the persistence of two single subunit bands (Supplementary Fig. 7).

**Toxic oligomers impair proteasome gate**. Substrates must pass through the gated translocation channel before gaining access to the proteolytic sites[44]. The A11+ oligomers are too large to enter the 13 Å translocation channel and directly inhibit β-subunit active sites; however, they could be impairing 20S proteasome function by impairing substrate entry through the gate or by allosterically impairing the active sites. To address this question, we used the α3ΔN proteasome mutant, which has a constitutively open gate[65]. If the oligomers impair proteasome activity by clogging the catalytic chamber or allosterically impairing the

active sites, then they should be able to impair the proteasome regardless if its substrate gate is in the opened or closed state. Alternatively, if the oligomers require a functioning gate for impairment, then they should not be able to impair a proteasome with a constitutively open gate, i.e., the α3ΔN 20S proteasome, which lacks only one of its seven α-subunit N-termini[44]. We added the three different A11+ oligomers: Aβ*56, α-Syn, and Htt-53Q, to the wild-type (WT) or the α3ΔN 20S proteasome and monitored substrate degradation. All three A11+ oligomers significantly impair WT proteasomes but do not impair the α3ΔN proteasomes (Fig. 3b–d). These results demonstrate that the A11+ oligomers require a functioning gate in order to impair the 20S proteasome. In addition, most active site proteasome inhibitors only inhibit one or two proteolytic sites, but the A11+ oligomers impair the degradation of substrates specific for each of the three different catalytic sites (Fig. 3b–d), further supporting a gating mechanism of impairment, since restricting substrate access would be expected to impair all types of substrates. Moreover, a translocation channel clogging mechanism can also be ruled out since the α3ΔN 20S proteasome could not be impaired. To further confirm an allosteric mechanism of proteasome impairment, we performed a substrate saturation curve

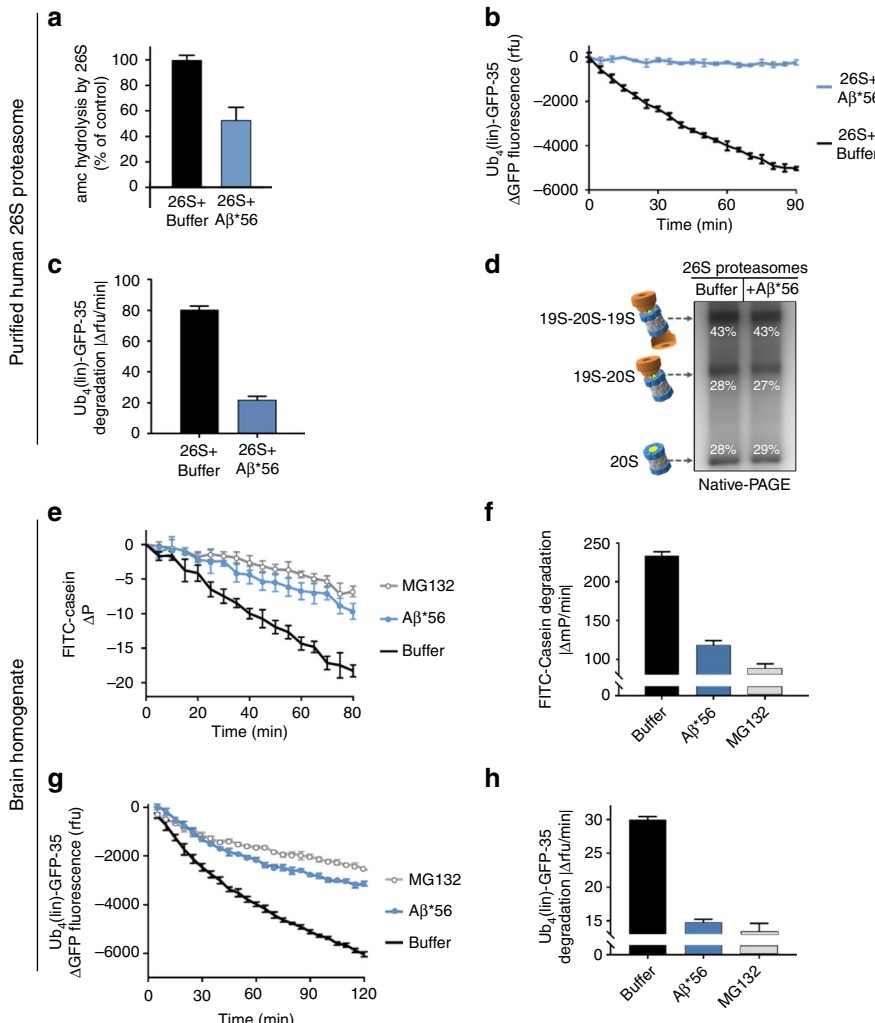

**Fig. 5** Aβ*56 oligomers inhibit ubiquitin-dependent and ubiquitin-independent degradation of full-length proteins. **a–d** Purified human 26S proteasomes. **a** Human 26S proteasome activity (LLVY-amc hydrolysis, rfu/min) with 2 μM Aβ*56 compared to buffer control. **b** Change in fluorescence of polyubiquitin-GFP fusion protein (Ub$_4$(lin)-GFP-35) in the presence purified human 26S proteasome, with and without 5 μM Aβ*56. **c** Rate of polyubiquitin-GFP fusion protein (Ub$_4$(lin)-GFP-35) degradation in **c** (Δrfu/min). **d** Purified human 26S proteasomes were incubated with or without Aβ*56 for 90 min at 37 °C, and separated by Native-PAGE and silver stained. Band density was quantified with ImageJ. Band density is shown as a percentage of total density of each lane. **e–h** Full-length protein degradation in mouse brain lysates. **e** Change in polarization of FITC labeled-casein in 2 μg mouse brain lysates, with and without 10 μM Aβ*56. **f** Rate of FITC-casein degradation in **e** (ΔmP/min). **g** Change in fluorescence of polyubiquitin-GFP fusion protein (Ub$_4$(lin)-GFP-35) in 2 μg mouse brain lysates, with and without 10 μM Aβ*56. **h** Rate of polyubiquitin-GFP fusion protein (Ub$_4$(lin)-GFP-35) degradation in **g** (Δrfu/min). The concentration of Aβ*56 oligomers is calculated based on Aβ monomeric peptide mass (4.5 kDa). All controls contained an equal volume of buffer identical to that of the Aβ*56 oligomers. The data are representative of three or more independent experiments performed in triplicate. Error bars represent ± standard deviation

on the WT 20S proteasome with and without Aβ*56 oligomers. We used nonlinear regression and the Michaelis–Menten equation to analyze the $K_D$ and $V_{max}$ of the two curves. We found that the Aβ*56 oligomers caused a decrease in the $V_{max}$ and an increase in the $K_m$ (Supplementary Fig. 8), which is consistent with allosteric inhibition (i.e., mixed inhibition—a form of non-competitive inhibition). Taken together, these data clearly demonstrate that all three diseases-related oligomers impair proteasome function by a similar allosteric mechanism, since all three A11+ oligomers require a closable gate on the 20S proteasome in order to impair it.

To validate the preparation of the open-gate α3ΔN 20S proteasomes, they were incubated with either a known gate-opening peptide (KANLQYYA[45] from the C-terminus of Rpt5, which includes the HbYX motif) or the β-subunit active site inhibitor, MG132. Treatment with MG132 completely inhibits

WT and open-gate α3ΔN proteasomes (Supplementary Fig. 9A) as expected for a pure proteasome preparation. The Rpt5 peptide increases WT 20S proteasome substrate degradation but failed to stimulate the open-gate α3ΔN 20S proteasome (Supplementary Fig. 9B) as expected for proteasomes with constitutively open gates. The preparations of pure α3ΔN 20S were approximately ten times more active than the WT 20S and thus ten times more WT 20S was used in these experiments to obtain comparable basal rates (Supplementary Fig. 9).

**Toxic oligomers stabilize the closed gate conformation.** Binding of the 19S ATPases C-termini HbYX motif into the 20S inter-subunit pockets induces a conformational change of the 20S α-subunits, which stabilizes the open state of the N-terminal gating residues[9, 45]. However, recent cryo-EM studies have highlighted the complexity of this gate-opening mechanism in the 26S

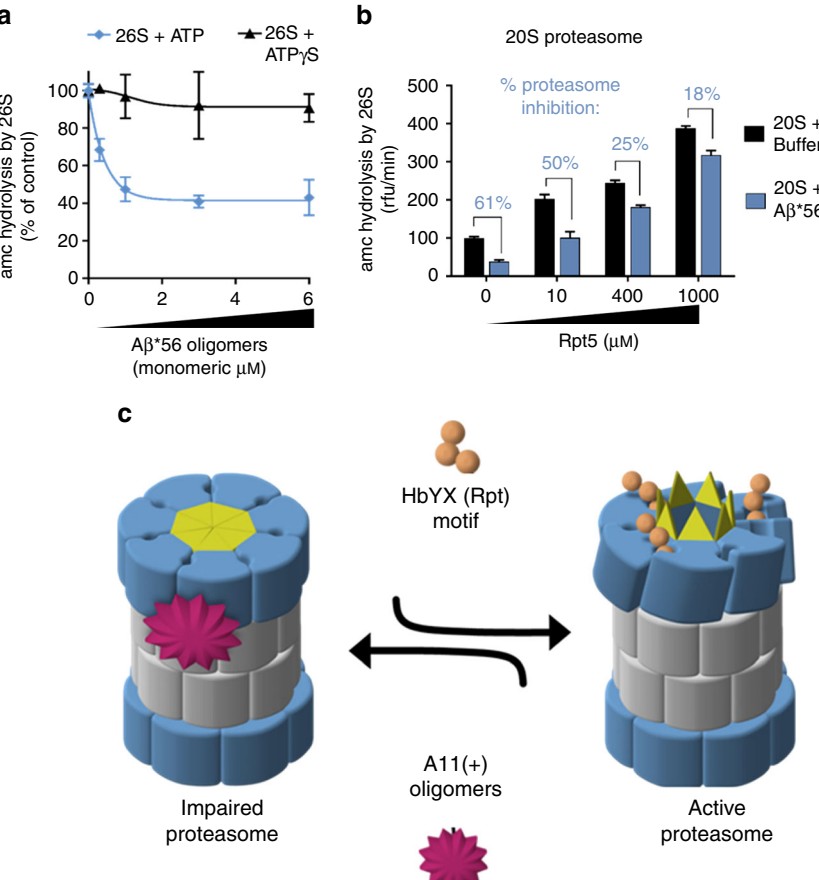

**Fig. 6** HbYX-dependent 20S gate opening counteracts inhibition by Aβ*56 oligomers. **a** Mammalian 26S proteasome activity (LLVY-amc hydrolysis, rfu/min) with 2 mM ATP or 10 μM ATPγS. Aβ*56 titration up to 6 μM. **b** 20S proteasome activity (nLPnLD-amc hydrolysis, rfu/min) with RPT5 peptide titration, with and without Aβ*56 (0.5 μM). **c** Schematic depicting our working model for proteasome inhibition by A11(+) oligomers (see text for details). The concentration of Aβ*56 oligomers is calculated based on Aβ monomeric peptide mass (4.5 kDa). All controls contained an equal volume of buffer identical to that of the Aβ*56 oligomers. The data are representative of three or more independent experiments performed in triplicate. Error bars represent ± standard deviation

proteasome when the 19S binds to a substrate (or when it is switched from an ATP-bound state to an ATPγS bound state)[66, 67]. It is less clear how the dynamics of the gate changes when the 19S associates with the 20S. Nevertheless, functional studies have shown that 19S binding to the 20S stimulates gate opening in the 20S proteasome[39, 40], in a HbYX motif dependent manner[47]. In contrast, the 11S family of proteasome activators (e.g., PA28α/β and PA26) bind to the 20S α-subunits and facilitate gate opening by a different mechanism. Although the 11S subunits also bind to the α-intersubunit pockets, they lack the HbYX motif and thus do not induce α-subunit conformational changes like the HbYX motif does. Instead, the 11S internal "activation loop" is required for gate opening. This "activation loop" directly contacts the base of the N-terminal gating residues and locally repositions them into the open conformation[68].

We hypothesized that binding of the A11+ oligomers to the 20S may specifically impair one of these distinct gate-opening mechanisms, which would provide evidence for the mechanism of oligomer-mediated proteasome impairment. If the oligomers could impair the PA26–20S complex, then it is expected that they would bind to the top of the 20S and compete with PA26 for binding to the 20S. Alternatively, if the oligomers do not compete with PA26 for binding to the 20S but they do impair the HbYX-dependent gate opening this indicates that oligomers must allosterically affect conformational changes that are caused upon HbYX motif binding. Another possibility is that the oligomers

affect both or neither mechanisms of gate opening. To assure that both ends of the 20S proteasome were bound by PA26 we used saturating amounts to stimulate 20S peptide degradation. None of the A11+ oligomers from Aβ, α-Syn, or Htt-53Q could impair the PA26–20S–PA26 complex (Fig. 4a–c). Thus, the PA26–20S–PA26 complex mirrors the results obtained for the α3ΔN 20S. To evaluate this possibility that the oligomers compete with PA26 for binding to the 20S, we generated a binding saturation curve for PA26 to 20S by monitoring 20S proteasome activation in the presence and absence of A11+ Aβ*56 oligomers. The apparent affinity of PA26 binding for the 20S did not decrease in the presence of the A11+ oligomers (Fig. 4d) and the oligomers did not impair PA26 at any concentration that was used, indicating that the oligomers do not compete with PA26 for binding to the 20S. In addition, the Aβ*56 oligomers could not impair PA26 mediated gate opening even at very high Aβ*56 concentrations (Fig. 4e). The PA28αβ proteasome activator from humans is a homologue of PA26 and thought to open the 20S gate in a similar activation loop-dependent manner[68]. Consistent with PA26 results, the A11+ Aβ*56 oligomers could not impair the human PA28αβ-mediated proteasome gate opening (Fig. 4f). Therefore, the A11+ oligomers bind to the 20S proteasome at a location separate from the 11S proteasome activators, PA26 and PA28αβ. Based on this we hypothesized that the oligomers stabilize the latent closed conformation of the α-subunits, which

is not affected by the PA26/28 activation loop-dependent gate opening.

To test this hypothesis, we asked if the A11+ oligomers could impair peptide and protein degradation by purified human 26S proteasomes. The A11+ Aβ*56 oligomers significantly impaired peptide degradation by the purified 26S proteasome compared to controls (Fig. 5a). To further test this possibility, we determined if oligomers could impair ubiquitin-dependent (Ub4(lin)-GFP-35) protein degradation by purified human 26S proteasomes. The Ub4(lin)-GFP-35 substrate we used to monitor ubiquitin-dependent degradation is a circularly permuted GFP with a linear tetra ubiquitin on N-terminus and a 35-residue unstructured region on the C-terminus that was created in the Matousheck lab. We found that Aβ*56 also strongly impaired the degradation of this structured protein (Fig. 5b, c) by the human 26S proteasome, which requires ATP-dependent unfolding and injection into the 20S core. These data suggest the oligomers impair the HbYX mechanism of gate opening. However, it is possible that the oligomer binding to the 20S could cause the 26S to disassemble into its 20S and 19S subcomplexes, which could also have the effect of impairing the ubiquitin-dependent protein degradation that we observed. To test this possibility, we incubated Aβ*56 oligomers and purified human 26S proteasome preparations together for 90 min at 37 °C before running the samples on native-PAGE (Fig. 5d). We quantified the silver stain band densities for isolated 20S, singly capped 26S, and doubly capped 26S. The relative ratio of these three populations of proteasomes did not change with the incubation with Aβ compared to control.

The prior experiments where done with highly purified components thus providing good cause and effect confidence for mechanistic analysis; however, the purified system cannot assess if the oligomers are able to bind to and impair the proteasome in an environment that more closely mimics a complex cellular environment. To address this, we prepared mouse brain lysates to determine if the toxic oligomers could still impair protein degradation by the proteasome in such a heterogeneous environment. We found that the brain lysates were highly competent to degrade the protein substrates FITC-casein (Fig. 5e, f) and Ub4 (lin)-GFP-35 proteins (Fig. 5g, h) similar to the purified 26S proteasome (Fig. 5b-c). We also assessed the proteasome activity component of this lysate by adding the proteasome inhibitor MG132, and found that the majority of the degradation activity, we observed was due to proteasome activity (Fig. 5e–h). When we assessed the degradation of these two-specific proteins in brain lysates, in the presence of the crosslink-stabilized Aβ*56, we observed extensive proteasome impairment—nearly as much as when MG132 was used (Fig. 5e–h). Therefore, the A11+ Aβ*56 oligomers retain enough specificity to bind to and nearly completely impair proteasome function even in a complex brain lysate. These data demonstrate that the oligomers do not disrupt the 26S complexes, and thus do not impair it by this mechanism, but must instead act, as we expected, on the gate. Impairment of the 26S proteasome also demonstrates that the oligomers must bind to the 20S even in the presence of the 19S, as was also observed for the PA26–20S complexes, demonstrating that the oligomers likely bind to the outer surface of the 20S proteasome (i.e., not on the gating surface) thus supporting the hypothesis that these toxic oligomers act allosterically preventing the 20S gate from opening properly. Since these results clearly demonstrate that A11+ oligomers are able to impair the 20S core particle by itself, then it is most likely that their impairment of the 26S proteasome is via the same mechanism, acting on the core particle. Importantly, this result demonstrates that A11+ oligomers can also impair proteasome function in a complex protein environment.

The 19S requires binding of ATP for it to bind to and induce gate opening in the 20S[69]. The 26S proteasome adopts multiple conformations during the ATP hydrolysis cycle and substrate degradation[66, 70–72]. In the presence of hydrolysable ATP, 26S proteasomes seems to alternate between active (open gate) and inactive (closed gate) states, with the inactive state predominating, and in contrast, non-hydrolyzing ATP analogues better stabilize the active (open gate) form of the proteasome[66, 73]. Interestingly, while the Aβ*56 oligomers impaired the 26S in the presence of ATP, they could not impair the 26S in the presence of the analog ATPγS (Fig. 6a). This shows that the oligomers are able to impair the normal physiological (with ATP) state of the 26S but not the synthetically opened state (using ATPγS), in which the open state is more "strongly" stabilized. We verified the integrity of the purified 26S proteasomes preparation via Native-PAGE to confirm that the observed activity came only from the 26S complexes (Supplementary Fig. 10) and not from any free 20S proteasome in the preparation. These results thus further support the hypothesis that these oligomers oppose the HbYX-dependent conformational changes that lead to gate opening.

To further test this hypothesis, we asked if the A11+ oligomers could block HbYX-dependent gate opening directly by the Rpt5 peptide (KANLQYYA), an established gate opening peptide[45], derived from the C-terminus of Rpt5. We added increasing concentrations of the Rpt5 peptide to the 20S proteasome with and without the A11+ Aβ*56 oligomers. In the absence of oligomers, the Rpt5 peptide significantly stimulated proteasome activity as expected. However, the oligomers impaired Rpt5 activation at all concentrations. Interestingly, the more Rpt5 was added the less effective the oligomers were to impair the proteasome (Fig. 6b). These results indicate that the oligomers impair HbYX-dependent gate opening, but also that HbYX peptide could overcome impairment by the oligomers at the higher concentrations (1 mM Rpt5 peptide was the highest concentration that could be tested due to its solubility). In contrast, the oligomers could not impair PA26 activation at any concentration of PA26. We interpret these results to mimic the ATP/ATPγS experiment (Fig. 6a), whereby the ATP state is a low HbYX occupancy state and the ATPγS state is a higher occupancy HbYX state. The rational is that ATP is rapidly hydrolyzed to ADP, and ADP cannot support HbYX-dependent gate opening. On the other hand, ATPγS is not hydrolyzed to ADP and thus it sustains the HbYX bound open-gate state or it could also enhance gate opening by other mechanisms[73]. These combined results fit well with a model, whereby the A11+ oligomers impair proteasome function by binding to the outer surface of the 20S barrel, and impair substrate entry by allosterically stabilizing the closed conformational state of the 20S α-subunits, in a way that directly counteracts the conformational changes that are required for HbYX-dependent gate opening.

## Discussion

The structural evolution of compartmentalized proteases was driven by the need to protect proteolytic activity from the cellular milieu, but still have the capacity to degrade select proteins in a regulated manner. The substrate-entry gate in the 20S proteasome thus plays a critical role in proteasome function and in cellular proteostasis. Here, we elucidate a common mechanism whereby soluble oligomers possessing a common 3D structure found in many neurodegenerative diseases potently inhibit 20S and 26S proteasome gate opening thus drastically impairing its function. While certain studies show some forms of aggregates do not impair the proteasome (which we also find Fig. 1b, d), the aggregates from these studies were not assayed for the presence of A11+ oligomers. Based on our results we proposed the following

mechanistic model (Fig. 6c) of how A11+ oligomers impair proteasome function: (1) A11+ oligomers bind with low nanomolar affinity (Figs. 1 and 2) to the outer surface of the α-subunits along the C2 axis (the presumed binding site); (2) by binding to this site the oligomers stabilize the closed conformation of the α-subunits and prevent spontaneous gate opening (Fig. 3); (3) activation loop-dependent gate opening (e.g., PA26) occurs normally in the presence of oligomers, since its mechanism only requires contact between the activation loops and the base of the gating residues (Fig. 4); (4) however, HbYX-dependent gate opening (e.g., the 19S regulatory particle or HbYX peptide) is inhibited as oligomer-bound α-subunits are unable to undergo the conformational changes required to open the gate (Figs. 5 and 6), which are stabilized by the bound oligomer. From a general mechanistic perspective, in this model one expects to observe opposing allosteric controls fought between two allosteric modulators that bind to distinct sites on the 20S proteasome. From this model, one expects to observe competition between two allosteric modulators (the HbYX motif and the oligomers) that bind to distinct sites on the 20S proteasome. In this sense, the HbYX motif is a positive allosteric modulator that induces gate opening, whereas the A11+ oligomers are negative allosteric modulators that induce gate closing. These diametrically opposed regulators thus fight to control the proteasome gate. Moreover, it appears that the HbYX mechanism is dominant since binding of the non-hydrolysable ATP analog, ATPγS, prevents inhibition by A11+ oligomers (Fig. 6a), though further confirmation is warranted.

These results demonstrate that oligomer-mediated impairment of proteasome function is not dependent on the sequence of the misfolded protein but rather the oligomer's 3D shape. Specifically, we found a consistent correlation between an oligomer's ability to impair the proteasome and recognition by the A11 antibody. While the physiological concentration of A11+ oligomers in neurons is unknown, if we consider that the affinity constant for the oligomers is low naomolar, and that the cellular concentration of the 20S is estimated to be low micromolar[74] then, with respect to this binding reaction, the 20S is saturating in the cell. This implies toxic oligomers will bind to the 20S irrespective of their cellular concentration, which begs the question: are the physiological levels of A11+ oligomers sufficient to impact protein degradation? Using laser capture microdissection and isolation of hippocampal pyramidal neurons from sporadic Alzheimer's Disease cases, Hashimoto et al.[75] determined the intraneuronal concentration of Aβ42 to be 3 μM, but what proportion of the intracellular Aβ42 is in oligomeric form is not known. Furthermore, Kisselev et al.[76] showed that the amount of proteasome inhibitor, Velcade™ that is used to treat multiply myeloma only inhibits protein degradation by about 10–25%. This result demonstrates that a relatively small alteration of protein breakdown can have a substantial impact on cell death. Consistent with this reasoning, stereotaxic unilateral infusion of lactacystin (a selective proteasome inhibitor) into the substantia nigra pars compacta of rats caused neurodegenerative disease like symptoms[22]. However, the percentage of proteasomes that must be active in neurons to maintain normal proteostasis is not known and thus we could only speculate about what level of intracellular A11 oligomers would be required to impact neuronal function. Nevertheless, as protein degradation begins to suffer as oligomers accumulate, the level of proteasome impairment is expected to increase exponentially as more proteins accumulate and oligomerize. Such a model would be expected to exhibit exponential progression kinetics, which coincides with the exponential deterioration that is observed over decades in most neurodegenerative diseases. These results build confidence that

such oligomers in neurons could impair proteasome function enough to contribute to the progression of these neurodegenerative diseases.

Future efforts are required to understand which structures within the A11 epitope facilitate 20S proteasome binding and impairment and if this phenomenon occurs in human disease conditions. Elucidation of this mechanism provides a compelling model to explain why proteasome function has been found to be impaired in virtually all neurodegenerative diseases. Interestingly, Choi et al.[77] showed that opening of the 20S proteasome gate in cells leads to enhanced cellular proteasome function, including ubiquitin-dependent protein degradation, decreased protein aggregates, and protection from oxidative stress. Our model provides a mechanistic framework to develop small molecules to counteract proteasome impairment via A11+ oligomers. Illustrating this potential mechanism of proteasome impairment identifies novel drug targets for developing small molecule activators of the proteasome gate. Such therapeutic interventions have the potential to restore proteostasis in patients suffering from neurodegenerative diseases.

## Methods

**Proteasome purifications**. Mammalian 20S proteasomes were isolated from bovine liver as described[78]. Briefly, cleared liver homogenate was passed over DE53 column. Protein was eluted with a stepwise NaCl gradient. Fractions with significant proteasome activity were pooled and further separated by a strong anion-exchange column (ResourceQ, GE Healthcare) eluting with NaCl gradient. Fractions with high suc-LLVY-amc hydrolysis were pooled for further purification using a hydroxyapatite column (CHT-I, Bio-Rad) and eluted by KPO₄ gradient. Fractions with high proteasome activity were pooled and further purified by SEC (S-400, GE Healthcare). Eluted fractions were pooled and purity of 20S proteasomes (>98%) was determined by SDS-PAGE and quantified by densitometry (ImageJ, NIH). Mammalian 26S proteasomes were isolated from rabbit muscle using the Ubl affinity purification as described[79]. Human 26S proteasomes were affinity purified on a streptavidin column from the HEK293-β4-biotin cell line as described[77]. Recombinant PA26 was expressed in BL21-STAR *Escherichia coli* and purified by affinity with a Ni-NTA column (Qiagen), as described[80]. Recombinant human PA28αβ was expressed in BL21-STAR *E. coli* and purified by affinity with a Ni-NTA column (Qiagen), as described[81]. WT and mutant α3ΔN yeast 20S proteasomes were expressed and purified by anion-exchange chromatography as described[82]. Fluorogenic substrate peptides were purchased from BostonBiochem (suc-LLVY-amc) and EZBiolabs (ac-nLPnLD-amc and ac-RLR-amc). Rpt5 peptides were synthesized by EZBiolabs. Protein concentrations were determined by Bradford assay (Thermo Scientific).

**Proteasome activity assays—peptide substrates**. Unless otherwise specified, bovine 20S (0.5 nM), rabbit muscle 26S (0.4 nM), yeast WT 20S (1.4 nM), or yeast α3ΔN 20S (0.14 nM) proteasomes were assayed using fluorogenic peptides, as described[45] in 96-well black flat bottom untreated plates (Costar). Briefly, proteasomes were incubated in a reaction buffer containing 50 mM Tris–HCl (pH 7.4), and 100 μM fluorogenic substrate (suc-LLVY-amc, ac-nLPnLD-amc) or 10 μM fluorogenic substrate (boc-LRR-amc). 20S proteasomes were treated with Rpt5, or with PA28αβ or PA26 to induced gate opening as indicated. Rabbit muscle 26S proteasomes were used in the presence of 1 mM DTT, 10 mM MgCl₂, and 100 μM of fluorogenic substrate (ac-nLPnLD-amc) with either 2 mM ATP (99%, Sigma) or 10 μM ATPγS (95%, Sigma). Fluorescence was measured every 55S for 120 min (ex/em: 380/460 nm). The rate of increase in fluorescence intensity is directly proportional to proteasome activity. For all experiments, an equal volume of the appropriate control buffer (identical to the aggregate/oligomer buffer that is described below) was used for controls. All molar concentrations of Aβ, α-Syn, and Htt-53Q are calculated based upon the monomeric protein concentration.

**Proteasome activity assays—protein substrates**. FITC-casein (0.08 μg, Sigma) and Ub₄(lin)-GFP-35 (0.08 μg, a kind gift from Dr. Andreas Matousheck) degradation assays were carried out in 50 μl reactions using 96-half-well non-binding surface treated black plates (Corning) at 37 °C. The GFP substrate was generated as described[83]. Proteasomes were added to the reactions (1 μg 20S, or 0.9 μg human 26S) in the presence or absence of Aβ*56 oligomers (10 μM) and fluorescence was measured at every 60 s for 90 min. The data shown are the mean of three reactions, with a five-point moving average, and error bars represent ± standard deviation. Degradation rates were determined by calculating the slope of a line fit to the first 30 min of activity.

**Aβ1–42**. Synthetic Aβ(1–42) was purchased from Selleckchem, Anaspec, and EZBiolabs. To remove preexisting aggregates, synthetic Aβ peptide was dissolved in 100% hexafluoroisopropanol (HFIP), and incubated at 37 °C for 2 h with shaking (500 r.p.m.). The HFIP was removed and the remaining peptide films were stored at −80 °C until use. Monomeric Aβ was obtained by dissolving synthetic peptide in 100% anhydrous dimethyl sulfoxide (DMSO) (Thermo Scientific) at 5 mM and diluted with phosphate-buffered saline (PBS) to a final concentration of 50 μM immediately prior to use. Crude Aβ aggregates were prepared as described[84]. Aβ*56 oligomers were generated similar to Barghorn et al.[59]. Briefly, HFIP-treated peptide films were resuspended in 100% anhydrous DMSO (5 mM) and bath sonicated for 20 min before further dilution (400 μM) with 20 mM NaPO$_4$ pH 7.4, 140 mM NaCl, 0.2% SDS. The 100 μM Aβ was incubated at 37 °C for 6 h, diluted to 100 μM with ddH$_2$O, incubated at 37 °C for 18 h, centrifuged for 10 min at 3000×$g$, and the supernatant containing Aβ*56 oligomers was removed and dialyzed against 5 mM NaPO$_4$ pH 7.4, 35 mM NaCl. Where indicated, Aβ*56 oligomers were CL (before dialysis) with 1 mM glutaraldehyde (EM grade, Thermo Scientific) for 2 h at room temperature. The reaction was quenched by the addition of 1 M Tris–HCl pH 8 (to a final concentration of 10 mM) and incubated for an additional 30 min. Aβ*56 oligomers were purified by SEC (Superose 12 10/30, GE Healthcare) and eluted as a single major peak. Each preparation of Aβ*56 was confirmed to be A11 + by dot blot analysis as described below. To generate Aβ-HMW A11+ oligomers, the second Aβ incubation at 100 μM was extended to 26 h. To generate Aβ A11+ protofibrils, the second Aβ incubation at 100 μM was extended to 50 h. The Aβ A11 + protofibrils eluted from the Superose 6 column in a single peak at the void volume and were confirmed to be >700 kDa by Native-PAGE. All buffers were filtered with 0.2 μm membranes immediately prior to use. All SEC experiments were performed on an ÄKTApurifier (GE Healthcare) at 4 °C with 5 mM NaPO$_4$ pH 7.4, 35 mM NaCl at a flow rate of 0.5 mL/min. Aβ concentration was calculated by UV absorption at 280 nm (molar extinction coefficient 1940 M/cm) and confirmed with Bradford protein concentration assay (Thermo Scientific).

**α-Syn protein**. Human WT α-Syn with N-terminal his-tag in pET28a vector was expressed and purified from BL21-STAR E. coli using a Ni-NTA column (Qiagen) followed by an anion-exchange chromatography (HiTrapQ, GE Healthcare). Pure α-Syn monomers were obtained by SEC (Superose 12 10/30, GE Healthcare) immediately prior to use. The purity of α-Syn monomers (>98%) was determined by SDS-PAGE and quantified by densitometry (ImageJ, NIH). Crude α-Syn aggregates/oligomers were generated by incubating monomeric α-Syn (3 mg/mL) in PBS (20 mM NaPO$_4$ pH 7.4, 140 mM NaCl) at 37 °C for 7 h. After oligomerization, the oligomers were separated from the remaining monomers by SEC (Superose 12, GE Healthcare) and verified A11+ by dot blot.

**Huntingtin protein**. GST-tagged huntingtin exon 1 constructs with a 53 polyglutamine repeat (GST-Htt-53Q) and a 20 polyglutamine repeat (GST-Htt-20Q). Protein was expressed and purified from BL21-STAR E. coli as described[85]. Briefly, the GST-fusion protein was cleaved with PreScission Protease (GE Healthcare) at 4 °C according to manufacturer protocol. The free Htt-53Q proteins were further purified by SEC (Superose 12 10/30, GE Healthcare) to obtain a monomeric population immediately prior to oligomerization. The purity of Htt-53Q monomers (>95%) was analyzed with SDS-PAGE quantified by densitometry (ImageJ, NIH). Oligomers were generated by incubating monomeric Htt-53Q (1 mg/mL) at 37 °C for 1 h. Due to the rapid formation of Htt-53Q oligomers, monomeric Htt-20Q (which oligomerized at a much slower rate) was used for the monomer assay in Fig. 1b.

**Crude aggregate fractionation**. Insoluble aggregates were removed from crude aggregate preparations by centrifugation at 10,000×$g$ for 10 min. The supernatant containing soluble oligomers was transferred to a fresh Eppendorf tube and the remaining pellet was gently resuspended in PBS. The pellet fraction was centrifuged twice more before final resuspension at 1 mg/mL in PBS. The fibrillar nature of the insoluble fraction was confirmed by Thioflavin-T (Sigma) fluorescence in comparison to monomer preparation controls as described below.

**SDS-PAGE and Native-PAGE**. Proteins were separated by SDS-PAGE using NuPAGE™ 4–12% Bis-Tris protein gels (Invitrogen), or separated by Native-PAGE using Novex™ 10–20% Tris-glycine or NuPAGE™ 3–8% Tris–acetate protein gels (Invitrogen), as indicated. Total protein was visualized with Coomassie stain (Simply Blue Safe Stain, Novex) or silver stain (Pierce Silver Stain kit, Thermo Scientific) as indicated according to manufacturer instructions. Immunoblots were performed as described below. Native-PAGE in-gel 26S proteasome activity assay was performed using NuPAGE™ 3–8% Tris–Acetate gels (Invitrogen). Samples were mixed with Novex™ Tris-glycine native sample buffer (×2) (Invitrogen) just before loading. Electrophoresis was carried out in Novex™ Tris-glycine native running buffer (Invitrogen) (with 0.5 mM DTT, 1 mM ATP, and 5 mM MgCl$_2$) at 4 °C and 150 V for 4 h. Native gels containing 26S proteasomes were incubated with reaction buffer (50 mM Tris pH 7.5, 10 mM MgCl$_2$, 2 mM ATP, 1 mM DTT, 50 μM suc-LLVY-AMC) for 30 min at 37 °C. Fluorescent bands around proteasomes were visualized by standard gel-imaging systems for DNA staining by ethidium bromide.

**Immunoblotting**. For Native-PAGE western blots, proteins were transferred to nitrocellulose membrane (GE) using Tris-glycine transfer buffer (Novex). Primary antibodies were purchased from Invitrogen (anti-oligomer A11, and anti-Aβ N-terminus clone NAB228) and diluted 1:1000 in TBST +5% nonfat milk prior to use. AlexaFluor-647 conjugated secondary antibodies (Invitrogen) were diluted 1:3500 in TBST prior to use. Membranes were blocked for 1 h at room temperature in TBST +10% nonfat milk, briefly washed with TBST, incubated with primary antibody for 1 h at room temperature, washed with TBST (3 × 5 min), incubated with secondary antibody for 1 h at room temperature, washed (3 × 5 min), and imaged on a Molecular Dynamics Typhoon 9410 Variable Mode Imager. Dot blots were performed by spotting protein on 0.1 μM nitrocellulose membranes and processed the same as Western blots.

**Antibody neutralization assays**. Anti-oligomer A11 (Invitrogen) and Aβ N-terminal antibody (clone NAB228, Invitrogen) were buffer exchanged to 50 mM Tris (pH 7.4) with Zeba spin desalting columns (Thermo Scientific). The antibodies (0.5 μg) were incubated with Aβ*56 (50 μM) or control buffer for 25 min at 37 °C before adding to proteasome activity assays.

**Crosslinking Aβ*56 and 20S proteasomes**. Mammalian 20S proteasomes were buffer exchanged to 10 mM NaPO$_4$ (pH 7 with Zeba spin desalting columns (Thermo Scientific) and incubated with Aβ*56 oligomers (or an equal volume of control buffer) for 45 min at 37 °C. One millimolar glutaraldehyde was added to the proteins, gently mixed, and incubated for 5 min at 37 °C. Crosslinking reactions were quenched by the additional 1 M Tris–HCl pH 8 (1 mM). Proteins were separated by SDS-PAGE and Native-PAGE and visualized with silver stain or immunoblotting as described above.

**Oligomer characterization**. For atomic force microcopy imaging, preformed Aβ-iO were deposited on freshly cleaved mica (Ted Pella Inc., Redding, CA) and allowed to sit for 30 s. The mica substrate was then washed with 200 μL of ultrapure water and dried with a gentle stream of nitrogen. Samples were imaged in tapping mode via ex situ AFM using a Nanoscope V MultiMode scanning probe microscope (Veeco, Santa Barbara, CA). AFM images were analyzed with Matlab equipped with the image processing toolbox (Mathworks, Natick, MA). For negative stain electron microscopy, 6 μl of preformed Aβ*56 oligomers were applied to ultra-thin copper 400 mesh carbon grids (Electron Microscopy Sciences) and imaged on a JEOL JEM-2100 TEM.

**Thioflavin-T florescence measurement**. ThT (Sigma) was dissolved (1 mM) in PBS, filtered through a 0.2 μM syringe, and stored at −20 °C until use. For the assays, 3 μg of Aβ was incubated at room temperature for 10 min in 100 μM of PBS with 20 μM ThT and fluorescence was measured (ex/em: 450/490 nm) in a Synergy2 plate reader (GenTek).

**Statistical analysis**. The data were analyzed using an unpaired Student's $t$-test (Prism). For all statistical analyses, a value of $p < 0.05$ was considered significant.

**Data availability**. The authors declare that the data supporting the findings of this study are available within the paper and its supplementary information files, and are available from the corresponding author upon request.

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

## Acknowledgements

We thank the members of the Smith Lab for the helpful and valuable discussions, and the protein core at WVU for their services. We thank Dr. Andreas Matouschek and his lab for generously providing us with the Ub4(lin)-GFP-35-His[6] plasmid, Dr. Olivier Coax for kindly providing the PA28αβ plasmid, and Dr. William Wonderlin for kindly providing the α-Synuclein plasmid. We thank Dr. Justin Legleiter for use of his AFM equipment and kindly providing the Htt-53Q and Htt-20Q constructs. This work was supported by NIH-R01GM107129 to D.M.S.

## Author contributions

T.A.T. designed, performed, and analyzed the various experiments in this manuscript (with input from D.M.S.). T.A.T. purified most proteins used in the manuscript, R.T.A. purified the Ub4(lin)-GFP-35 substrate. T.A.T. and R.T.A. performed the α-Synuclein experiments. Results were analyzed and interpreted by T.A.T., R.T.A., and D.M.S. Manuscript preparation was done by T.A.T. and D.M.S. All authors reviewed the results and approved the final version of this manuscript.

## Additional information

**Competing interests:** The authors declare no competing interests.

