## [Peer Review File(PDF 440 kb) · Nature Communications]

Reviewers' comments:

Reviewer #1 (Remarks to the Author):

In this study, Thibaudeau and co-workers investigate the in vitro effect of protein aggregates, consisting of three different types of aggregation-prone proteins involved in neurodegenerative diseases, on the enzymatic activity of the proteasome. They show that oligomers and not monomers or fibrillar aggregates impair the proteolytic activity of the proteasome. They provide evidence that indicate that the oligomers impair the gate opening of the proteasome and thereby restrict access of the substrates to the proteolytic chamber. The experimental work is sound although the manuscript suffers from overstatements and is at parts sloppy (mostly the references to the figures). The experimental design and the approaches that the authors are using are very similar to an earlier study in which the inhibitory effect of prion oligomers on proteasomes was studied. Although the authors cite this paper and must be aware of it, they do not discuss the similarities of the studies even though they both come to the same conclusion based on very similar experimental data. In the light of this earlier study, the novelty of this work lies in the observation that also three other aggregation-prone proteins in their oligomeric state inhibit the proteasome by blocking the opening of the gate. I find this study interesting but do not feel that it gives the conceptual advance that one would expect of a publication in Nature Communications.

Major concerns

- The authors do not put their finding in relation to an earlier study published by Deriziotis et al in the EMBO Journal in 2011 that used a very similar approach to decipher the inhibitory effect of prion oligomers on proteasomal degradation. In this earlier study, they also used an in vitro system with wild-type and open-gate proteasomes and purified oligomers and came to very similar conclusions.
- While it is true that impairment of the ubiquitin/proteasome system causes neurodegeneration in in vivo models, it is still questioned if inhibition of the ubiquitin/proteasome system is part of the etiology of neurodegenerative disorders. A number of studies have shown that the ubiquitin/proteasome system is functional in several mouse models for neurodegenerative diseases. This does not necessarily mean that there is no inhibition of the proteasome but also when it comes to this point the literature is far from clear with groups reported a decrease, no change or even an increase in proteasome activity. The literature should be presented in a more balanced way.
- This study heavily relies on in vitro approaches with artificial substrates and proteasomes. Even the final experiments in which the inhibitory effect is shown in brain lysates, the assay still relies

on artificial substrates. These type of in vitro experiments typically show that something can happen and not necessarily that this also does happen in the physiological context of the disease. Since it is not clear from the literature whether proteasome activity is reduced in neurodegenerative diseases, the authors should be careful in extrapolating these findings to the disease state.

- I am aware of the fact that in most experiments it is not possible to include as a control a non-aggregated protein but at least in Fig 1A, it would be helpful if the wild-type huntingtin would have been included as a control.

- When comparing monomers, oligomers and aggregates, the molar concentrations may sometimes be ambiguous. I assume that the authors are referring to the molar concentrations of the monomer but in case of the oligomers, intermediate oligomers and aggregates, this means that the molar concentrations of the active unit are much lower as they are composed of multiple monomers. Did the authors take this in consideration?

- Fig 3A: Does monomer also bind to the proteasome? It would be good to include this as a control.

Minor concerns

- Introduction: Is PINK really a component of the ubiquitin/proteasome system? It phosphorylates Parkin and ubiquitin but it is a bit of a stretch in my opinion to classify it as a component.

- Huntingtin exon 1 is abbreviated as HDEx1. I think it would be better to abbreviate Huntingtin as Htt. HD refers to Huntington's disease, which is the disease and not the protein.

- Fig 1F is mentioned in the text but not shown.

- Fig 2 at the end of the paragraph should be Fig. 2D.

- The second time that Fig 2J is mentioned it should be Fig 2K.

- Fig 2K should be Fig 2L.

- Fig S7A should be Fig S8B.

- Fig S7B should be Fig S8A.

- Fig 4G is shown but not mentioned in the text and does not have a figure legend.

- Fig S8 should be Fig S9.

- In the last part "Toxic oligomers impair 26S activity in brain lysates" there is no reference to any of the figures.

- Conclusions: "Dementional structures" should be "dimensional structures".

- It is stated that the oligomers induce gate closing. Can this really be concluded? I would rather say that the oligomers prevent gate opening.

Reviewer #2 (Remarks to the Author):

Summary: The authors investigate how the proteasome pathway is inhibited by aggregation-prone proteins in neurodegenerative diseases. The introduction contains sufficient background information and the order of aspects and experiments is reasonable/logically presented. All figures are clear and easy to read. The authors begin with a simple but effective experiment comparing the impact of monomers, soluble and insoluble fractions of A β , alpha-Synuclein and HD-x53Q on proteasomal activity. Only soluble aggregates, that are A11+ inhibited the proteasome activity remarkably. After that they focused on A β *56 (crosslinked?) oligomers to investigate step by step the possible mechanism of this inhibition. The findings that inhibition of the proteasome is dependent on the three-dimensional structure and correlates with A11+ might be important for further studies focusing on drug development. In summary the presented data is well structured and adds new findings to the current knowledge, it is interesting and it meets the criteria for publication in the journal.

Minor comments:

1. The authors have to carefully check the figure legends with their content and how they are mentioned in the text. For example, Figure S9 is never mentioned in the text. From the text content of S9 is described as S8. Another example is Figure 2 F to L. Figure 2E was divided into left and right but the right panel is then marked as F. This makes the article harder to read and to understand.

Figure 1B colors of bars in particular soluble versus insoluble should be changed to easier discriminate them from each other. Same is true for Figure 5E-H.

2. The authors should distinguish between A β *56 oligomers and A β *56 crosslinked oligomers (e.g. A β *56c). It is not clear, if all experiments were performed with a crosslinked version of A β *56. Otherwise they may add a sentence like “for all following experiments was used”.

3. Some minor spell checking like “dementional”.

4. Figure 6A shows purified human 26S proteasomes, but how these were obtained lacks in M+M section.

5. The discussion may benefit by some consideration about the physiological relevance of the concentrations of a particular oligomer used to inhibit the proteasome?

6. “Aggregation prone proteins are dynamic” They are not just dynamic, they are also polymorphic which is an unusual property for a protein. There are a lot of dynamic proteins that are not aggregation prone. More aggregation prone proteins are natively unfolded or unstructured, which is not the same as dynamic.

7. A11 is a polyclonal serum, not a single monoclonal antibody.

Reviewer #3 (Remarks to the Author):

The abstract of Thibaut et al. provides an accurate description of the MS content, and does not need reprising here.

It is widely recognized that neurological diseases characterized by protein aggregate formation or long polyglutamine repeats are associated with defects of the ubiquitin proteasome system (UPS). What is less clear and the subject of long-standing discussion is whether the protein defects are causal of or are merely associated with UPS defects, and, if causal, the mechanism involved. The present MS provides data indicating that three different proteins associated with neurologic disease can impair in vitro proteasome function but (and this is seemingly novel) they must form oligomers to do so. Higher order aggregates or monomers are inactive, but correctly sized oligomers that are isolated by size-fractionation or genetically engineered are active. The activity used to test for inhibition by the oligomers is cleavage by 20S proteasomes of fluorogenic peptides. Each of the three 20S distinct proteolytic sites was tested with a distinct and specific fluorogenic peptide. Each of the three showed inhibition with the three disease-associated oligomers.

The assay chosen provides two sources of skepticism, not necessarily disabling. 1. For degradation of native proteins (but not those which are weakly structured) the 19S regulatory complex equipped with an ATPase ring and degron recognition elements must be present in association with 20S. 2. The native activity of the proteasome is as a protease, not a peptidase. Together these two raise a question as to whether the observations reported are relevant to disease pathogenesis. The finds described may be relevant to pathogenic processes, but this remains uncertain. It would be of value to test whether 26S proteasomes (20S plus 19S) engaged in degrading a protein substrate are inhibited by the oligomers that inhibit 20S peptidase activity.

The authors favor an interpretation of their data whereby "... these oligomers inhibit the 20S proteasome by allosterically inhibiting the opening of the substrate-gate in the 20S core particle, preventing the 19S regulatory particle from injecting substrates into the degradation chamber."

This dual claim (allostery, gate opening) can be questioned on two grounds.

Allosteric regulation implies that the regulator (oligomers) acts on a site distinct from that/those engaged by a substrate. An alternative interpretation of the present data is that the oligomers are competitive inhibitors, which share some common site or sites of engagement with the substrate peptide. Such a common site could perhaps be at gate association or passage. The argument made against this interpretation is that open gate mutants of 20S do not display inhibition of peptidase activity by the oligomers. But open gate mutants have much higher throughput of peptides (the MS should tell us how much higher; but it does not). Competitive inhibition requires competition at some rate-determining step. A gating-related process may be rate-determining in wild type, but not in the mutant. Hence the findings with the mutant proteasomes (no inhibition observed) do not exclude competitive inhibition in the wild type case. As described in biochemistry texts, testing for competitive inhibition requires using several non-

saturating amounts of substrate (here peptides) and several concentrations of inhibitor. A double reciprocal plot shows a characteristic pattern for competitive inhibition. This experiment should be done.

The second technical question is related to the adequacy of the evidence for gate opening. Peptidase activity is commonly and conveniently used, but this surrogate for structure does not offer definitive evidence. That requires structural data. For 20S, negative stain EM is likely to suffice, but cryoEM would provide the higher resolution data that might be required to show partial or intermittent effects of inhibitor plus peptide on gate status. EM would suffice to answer the question for 20S. The claim that oligomers alter the ability of 26S to open the gate is more technically demanding, but also well within the capacity of cryoEM.

Some minor points-

Related to fig. 1. Do A11+ oligomers maintain oligomeric structure with antibody? Do they remain soluble with antibody?

Various figures. Absolute values of 100% peptidase activity (no inhibitor) are shown for WT and gate mutant 20S. What are the absolute values on some common scale of activity?

Data described as in sup. Fig. 7A are actually in figure 8.

For the section headed "Toxic Oligomers impair 26S activity in brain lysates", the data referred to in this section seem to be missing in the MS provided for review.

Responses to Reviewers' comments:

Reviewer #1 (Remarks to the Author):

In this study, Thibaudeau and co-workers investigate the in vitro effect of protein aggregates, consisting of three different types of aggregation-prone proteins involved in neurodegenerative diseases, on the enzymatic activity of the proteasome. They show that oligomers and not monomers or fibrillar aggregates impair the proteolytic activity of the proteasome. They provide evidence that indicate that the oligomers impair the gate opening of the proteasome and thereby restrict access of the substrates to the proteolytic chamber. The experimental work is sound although the manuscript suffers from overstatements and is at parts sloppy (mostly the references to the figures). The experimental design and the approaches that the authors are using are very similar to an earlier study in which the inhibitory effect of prion oligomers on proteasomes was studied. Although the authors cite this paper and must be aware of it, they do not discuss the similarities of the studies even though they both come to the same conclusion based on very similar experimental data. In the light of this earlier study, the novelty of this work lies in the observation that also three other aggregation-prone proteins in their oligomeric state inhibit the proteasome by blocking the opening of the gate. I find this study interesting but do not feel that it gives the conceptual advance that one would expect of a publication in Nature Communications.

Major concerns

- The authors do not put their finding in relation to an earlier study published by Deriziotis et al in the EMBO Journal in 2011 that used a very similar approach to decipher the inhibitory effect of prion oligomers on proteasomal degradation. In this earlier study, they also used an in vitro system with wild-type and open-gate proteasomes and purified oligomers and came to very similar conclusions.

Indeed, we are aware of the Deriziotis et. al. paper, as the corresponding author of this manuscript is a co-first author of the Deriziotis et al. paper. This Deriziotis study had shown (as many prior studies have) that a misfolded and aggregated protein could impair the proteasome, which was not novel. Specifically, this study was focused on soluble aggregates of recombinant mouse PrP^{Sc}. The primary contribution was that it showed that aggregates of PrP^{Sc} could impair the proteasome, and that it did so by affecting the gating mechanism. However, the fact that the proteasome impairing A11+ oligomers identified here shares a mechanism of impairment with PrP^{Sc} aggregates, does not preclude this study from providing a conceptual advance. We argue that the advancement this study brings to the field is the identification of a specific oligomeric conformation found in Alzheimer's, Parkinson's, and Huntington's disease that substantially impairs proteasome function in a way that could contribute to all of these neurodegenerative diseases. Creutzfeldt–Jakob disease (caused by PrP^{Sc} misfolding) is very rare, with an occurrence of one in every one million individuals per year. In contrast, Alzheimer's disease alone occurs in 50% of individuals 85+ years of age. Moreover, the Deriziotis study did not use purified oligomers, as suggested by the reviewer, (they were quite heterogeneous PrP^{Sc} aggregates) and the open-gate proteasomes that were used were from yeast. In our study, we purified, to homogeneity, a specific conformation of a pathological oligomer, identified its conformational status, and carefully and extensively characterized its mechanism

of impairment on purified human (and bovine) proteasomes. We even went as far as to show impairment of ubiquitin-dependent protein degradation in brain lysates. Therefore, our characterization of the mechanism of impairment was far more biochemically rigorous and relevant to human disease than the Deriziotis study. The experiments and discussion of mechanisms is ancillary to the importance of identifying the oligomer conformation responsible for proteasome impairment, which the Deriziotis study simply did not do. **All this being said, we have added the below sentence to the manuscript to ensure that this study is understood in the context of the prior Deriziotis study as suggested by the reviewer.**

Added manuscript text: *“A study by Deriziotis et al. (2011)³⁶ showed that heterogeneous aggregates of mouse prion protein can impair the proteasome via a gating mechanism. In our study, we purified to homogeneity a specific conformation of a pathological oligomer, identified its conformational status, and carefully and extensively characterize its mechanism of impairment on human and mammalian 20S/26S proteasomes. The advancement our study brings to the field is the identification of a specific oligomeric conformation found in Alzheimer’s, Parkinson’s, and Huntington’s diseases that substantially impairs proteasome function in a way that may contribute to all of these neurodegenerative diseases.”*

- While it is true that impairment of the ubiquitin/proteasome system causes neurodegeneration in in vivo models, it is still questioned if inhibition of the ubiquitin/proteasome system is part of the etiology of neurodegenerative disorders. A number of studies have shown that the ubiquitin/proteasome system is functional in several mouse models for neurodegenerative diseases. This does not necessarily mean that there is no inhibition of the proteasome but also when it comes to this point the literature is far from clear with groups reported a decrease, no change or even an increase in proteasome activity. The literature should be presented in a more balanced way.

We made changes to the introductory text to balance the presented literature.

Added manuscript text: *“Decreased proteasome function has been reported in a broad array of chronic neurodegenerative diseases. Impaired proteasome function has been implicated, as a primary cause or a secondary consequence, in the in the pathogenesis of many neurodegenerative diseases^{2, 17-21.}” and “However, the vast majority of neurodegeneration is idiopathic in origin and the involvement of the UPS is less clear¹⁷. What is clear in these diseases is that proteins that are normally degraded are not properly degraded after misfolding occurs, leading to their accumulation.”*

- This study heavily relies on in vitro approaches with artificial substrates and proteasomes. Even the final experiments in which the inhibitory effect is shown in brain lysates, the assay still relies on artificial substrates. These types of in vitro experiments typically show that something can happen and not necessarily that this also does happen in the physiological context of the disease. Since it is not clear from the literature whether proteasome activity is reduced in neurodegenerative diseases, the authors should be careful in extrapolating these findings to the disease state.

Based on this comment we made changes to address this point in the discussion. For example, *“Future efforts are required to understand which structures within the A11 epitope facilitate 20S proteasome binding and impairment and if this phenomenon occurs in human disease conditions.”* Minor changes were made to the last discussion paragraph as well.

- I am aware of the fact that in most experiments it is not possible to include as a control a non-aggregated protein but at least in Fig 1A, it would be helpful if the wild-type huntingtin would have been included as a control.

Indeed we used monomeric A β and α -synuclein in Figure 1B. Monomeric huntingtin-20Q (non-pathogenic glutamine repeat) was used as the monomer control for huntingtin-53Q (labeled in Figure 1B and the figure legend). In addition, the Materials and Methods section states: “Due to the rapid oligomerization of huntingtin-53Q, we used huntingtin-20Q (wild-type) as the monomer control”. Based on this comment we have increased the size of “Htt-20Q” text label in Figure 1B bar graph to avoid confusion.

- When comparing monomers, oligomers and aggregates, the molar concentrations may sometimes be ambiguous. I assume that the authors are referring to the molar concentrations of the monomer but in case of the oligomers, intermediate oligomers and aggregates, this means that the molar concentrations of the active unit are much lower as they are composed of multiple monomers. Did the authors take this in consideration?

This is a good point and we did take it into consideration. To increase clarity the figure legend for Figure 1 now reads: “The concentrations of aggregates are calculated based on the monomeric peptide or protein mass (A β , 4.5 kDa; α -Syn, 14 kDa; and Htt-53Q, 22 kDa).” Figures legends 2-6 state: “Concentration of aggregates are calculated based on the mass of peptide monomer”. The methods for “proteasome assays” state: “All molar concentrations of A β , α -Syn, and Htt-53Q are calculated based upon the monomeric protein concentration.” Figure 2C shows the IC₅₀ value calculated for [A β *56 dodecamer], and the x-axis is labeled [A β monomer]. If desired, we can also show [A β *56 dodecamer] in addition [A β monomer] for all A β *56 experiments. The huntingtin-53Q and α -Syn oligomers were not homogenous in size as determined by Native-PAGE, therefore we cannot confidently calculate [oligomer] for these proteins. We believe these changes address this concern.

- Fig 3A: Does monomer also bind to the proteasome? It would be good to include this as a control.

Monomers are small unstructured proteins which are readily degraded by the proteasome. To be degraded they must bind to the proteasome at the active sites. However, no evidence has been found that they bind to sites outside of the active sites, and our data here shows that they do not affect proteasome activity.

Minor concerns

- Introduction: Is PINK really a component of the ubiquitin/proteasome system? It phosphorylates Parkin and ubiquitin but it is a bit of a stretch in my opinion to classify it as a component.

The text did not claim that PINK is a “component” of the UPS, but that PINK “affects” the UPS, which, as the reviewer indicates, it does. From the manuscript: “mutations...many of which affect components of the UPS (e.g. PARK1, PINK)”.

- Huntingtin exon 1 is abbreviated as HDEx1. I think it would be better to abbreviate Huntingtin as Htt. HD refers to Huntington's disease, which is the disease and not the protein.

We agree, and the changes have been made throughout the text and figures.

- Fig 1F is mentioned in the text but not shown.

- Fig 2 at the end of the paragraph should be Fig. 2D.

- The second time that Fig 2J is mentioned it should be Fig 2K.

- Fig 2K should be Fig 2L.

- Fig S7A should be Fig S8B.

- Fig S7B should be Fig S8A.

- Fig 4G is shown but not mentioned in the text and does not have a figure legend.

- Fig S8 should be Fig S9.

- In the last part "Toxic oligomers impair 26S activity in brain lysates" there is no reference to any of the figures.

We regret and apologize for these confusing mistakes and take responsibility for them. They are now fixed in the current manuscript. Thank you for bringing these to our attention.

- Conclusions: "Dementional structures" should be "dimensional structures".

Correction made.

- It is stated that the oligomers induce gate closing. Can this really be concluded? I would rather say that the oligomers prevent gate opening.

We understand the reviewer's perspective, and gave it considerable thought and discussion. Though it is a nuanced point, it is relevant and could be mechanistically distinguished, though this is very difficult to do. However, we would argue that HbYX containing peptides are able to induce gate-opening, and when they are combined with inhibitory oligomers, the effects of gate-opening are reversed and thus, oligomers do appear to induce gate closing. In addition, binding of the 19S to the 20S forms the 26S and activates peptidase activity through gate-opening, and oligomers are also able to impair 26S proteasome. So from these two perspectives, we believe that it is relevant to indicate that the oligomers can induce gate closing.

Reviewer #2 (Remarks to the Author):

Summary: The authors investigate how the proteasome pathway is inhibited by aggregation-prone proteins in neurodegenerative diseases. The introduction contains sufficient background information and the order of aspects and experiments is reasonable/logically presented. All figures are clear and easy to read. The authors begin with a simple but effective experiment comparing the impact of monomers, soluble and insoluble fractions of Abeta, alpha-Synuclein and HD-x53Q on proteasomal activity. Only soluble aggregates, that are A11+ inhibited the proteasome activity remarkably. After that they focused

on A β *56 (crosslinked?) oligomers to investigate step by step the possible mechanism of this inhibition. The findings that inhibition of the proteasome is dependent on the three-dimensional structure and correlates with A11+ might be important for further studies focusing on drug development. In summary the presented data is well structured and adds new findings to the current knowledge, it is interesting and it meets the criteria for publication in the journal.

Minor comments:

1. The authors have to carefully check the figure legends with their content and how they are mentioned in the text. For example, Figure S9 is never mentioned in the text. From the text content of S9 is described as S8. Another example is Figure 2 F to L. Figure 2E was divided into left and right but the right panel is then marked as F. This makes the article harder to read and to understand. We regret and apologize for these confusing mistakes. They are now fixed in the current manuscript. Thank you for bringing these to our attention.

Figure 1B colors of bars in particular soluble versus insoluble should be changed to easier discriminate them from each other. Same is true for Figure 5E-H.

Corrections made. We altered the figure colors so the experimental groups are easier to discriminate when viewed in black and white print.

2. The authors should distinguish between A β *56 oligomers and A β *56 crosslinked oligomers (e.g. A β *56c). It is not clear, if all experiments were performed with a crosslinked version of A β *56. Otherwise they may add a sentence like “for all following experiments was used”.

We agree. We inserted “All following experiments utilize crosslinked A β *56 oligomers unless indicated otherwise.” to the end of Figure 2 legend and “These crosslinked oligomers are therefore ideally suited for further mechanistic and biochemical analysis to understand how oligomers impair proteasome function and are used in all following experiments unless stated otherwise.” into the main text.

3. Some minor spell checking like “dementional”.

Correction(s) made.

4. Figure 6A shows purified human 26S proteasomes, but how these were obtained lacks in M+M section.

Correction made to the Materials and Methods section: “Human 26S proteasomes were purified from the HEK293- β 4-biotin cell line as described³”

5. The discussion may benefit by some consideration about the physiological relevance of the concentrations of a particular oligomer used to inhibit the proteasome?

We appreciate this point and had considered addressing this in the discussion but opted to exclude it because it is speculative. There isn't any published evidence regarding the concentration of oligomers (much less the concentration of A11+ oligomers) within neurons from diseased brains. Instead, in the

discussion, we focus on the low nM affinity of oligomers for the proteasome. We would like the reviewer to consider that the affinity constant for the oligomers is low nM, and since the cellular concentration of the 20S is estimated to be micromolar (Russell et. al. JBC 1999), then the 20S is saturating in this binding reaction, and this implies that any oligomer that is generated will (based on Michaelis-Menten Kinetics) bind to the 20S irrespective of the oligomer's concentration in the cell. Our experiment showing proteasome inhibition in brain lysates also corroborate this conclusion. Therefore, to address the reviewers concern we have added the following sentences to the discussion: *“Moreover, while the physiological concentration of A11+ oligomers in neurons is unknown, if we consider that the affinity constant for the oligomers is low nM, and that the cellular concentration of the 20S is estimated to be micromolar (Russell et. al. JBC 1999), then, with respect to this binding reaction, the 20S is saturating in the cell. This implies that these toxic oligomers will bind to the 20S irrespective of their cellular concentration.”*

6. *“Aggregation prone proteins are dynamic”* They are not just dynamic, they are also **polymorphic** which is an unusual property for a protein. There are a lot of dynamic proteins that are not aggregation prone. More aggregation prone proteins are natively unfolded or unstructured, which is not the same as dynamic. We agree that the phrase *“Aggregation prone proteins are dynamic”* should be clarified. The text now reads: *“Aggregation-prone proteins have the unique property of conformational polymorphism. During amyloid formation, a variety of aggregate species are formed, ranging from small dimers to large insoluble fibrils.”* We appreciate the suggestion.

7. *A11 is a polyclonal serum, not a single monoclonal antibody.*

We purchase protein A purified anti-oligomer A11 antibody from Invitrogen. We apologize for not clarifying the polyclonal nature of the antibody in the text. Correction made: *“Kayed et al. (2003)⁵⁴ generated a polyclonal anti-oligomer antibody (A11)...”*.

Reviewer #3 (Remarks to the Author):

The abstract of Thibadeau et al. provides an accurate description of the MS content, and does not need reprising here.

It is widely recognized that neurological diseases characterized by protein aggregate formation or long polyglutamine repeats are associated with defects of the ubiquitin proteasome system (UPS). What is less clear and the subject of long-standing discussion is whether the protein defects are causal of or are merely associated with UPS defects, and, if causal, the mechanism involved. The present MS provides data indicating that three different proteins associated with neurologic disease can impair in vitro proteasome function but (and this is seemingly novel) they must form oligomers to do so. Higher order aggregates or monomers are inactive, but correctly sized oligomers that are isolated by size-fractionation or genetically engineered are active. The activity used to test for inhibition by the oligomers is cleavage by 20S

proteasomes of fluorogenic peptides. Each of the three 20S distinct proteolytic sites was tested with a distinct and specific fluorogenic peptide. Each of the three showed inhibition with the three disease-associated oligomers.

The assay chosen provides two sources of skepticism, not necessarily disabling.

1. For degradation of native proteins (but not those which are weakly structured) the 19S regulatory complex equipped with an ATPase ring and degron recognition elements must be present in association with 20S.

This point is the first of two points of skepticism, but it seems to simply be a statement of fact rather than a point of skepticism. We think the reviewer's intention for this point was to introduce the "source of skepticism" in point 2 below.

2. The native activity of the proteasome is as a protease, not a peptidase. Together these two raise a question as to whether the observations reported are relevant to disease pathogenesis. The finds described may be relevant to pathogenic processes, but this remains uncertain. It would be of value to test whether 26S proteasomes (20S plus 19S) engaged in degrading a protein substrate are inhibited by the oligomers that inhibit 20S peptidase activity.

Points 1-2. The reviewer is absolutely correct. To the reviewer's point, however, we presented ubiquitin-dependent 26S proteasome degradation of structured and unstructured proteins in Figure 5 and peptide degradation by the 26S proteasome in Figure 6 in the manuscript. Perhaps this data was accidentally missed by the reviewer. Nevertheless, to ensure flow/clarity and guard against this important data being missed by others, we have made minor changes to the order in which we present these data in the new manuscript. We believe this will help. Figure 5 now shows peptide and ubiquitin-dependent degradation by purified 26S proteasomes together in a single subsection (5A-C), followed by second subsection (5E-H) containing full-length protein degradation (ubiquitin-dependent and -independent) in mouse brain lysates. Figure 6 now contains data evaluating the oligomer mediated proteasome impairment and the HbYX mechanism of gate-opening. It is our hope that this small rearrangement improves the readability and flow of the manuscript. These data represent the data requested by the reviewer, and we believe they satisfy this concern.

The authors favor an interpretation of their data whereby "... these oligomers inhibit the 20S proteasome by allosterically inhibiting the opening of the substrate-gate in the 20S core particle, preventing the 19S regulatory particle from injecting substrates into the degradation chamber." This dual claim (allostery, gate opening) can be questioned on two grounds. Allosteric regulation implies that the regulator (oligomers) acts on a site distinct from that/those engaged by a substrate. An alternative interpretation of the present data is that the oligomers are competitive inhibitors, which share some common site or sites of engagement with the substrate peptide. Such a common site could perhaps be at gate association or passage. The argument made against this interpretation is that open gate mutants of 20S do not display inhibition of peptidase activity by the oligomers. But open gate mutants have much higher throughput of peptides (the MS should tell us how much higher; but it does not).

We now include the raw activity of wild type and mutant proteasomes on a common scale of absolute activity in **supplementary figure 9** to address this point. In short, the $\alpha 3\Delta N$ 20S preparations we used had approximately 10 times more specific activity than the WT 20S. WT and $\alpha 3\Delta N$ 20S proteasomes were assayed at the same time in the same 96 well plate. Due to the higher basal activity of the $\alpha 3\Delta N$ 20S proteasome, ten times less $\alpha 3\Delta N$ 20S (0.14nM) was used compared to the WT proteasome (1.4nM) to maintain comparable rate data for both reactions. In addition, separate experiments showed that oligomers also could not inhibit the $\alpha 3\Delta N$ 20S at comparable concentrations (e.g. 1.4nM) to the WT 20S. The concentrations of proteasomes used is notated in the figure labels for clarity. We also added to the text to increase clarity: *“The preparations of pure $\alpha 3\Delta N$ 20S were approximately 10 times more active than the WT 20S and thus ten times more WT 20S was used in these experiments to obtain comparable basal rates (Fig. S9).”*

Competitive inhibition requires competition at some rate-determining step. A gating-related process may be rate-determining in wild type, but not in the mutant. Hence the findings with the mutant proteasomes (no inhibition observed) do not exclude competitive inhibition in the wild type case. As described in biochemistry texts, testing for competitive inhibition requires using several non-saturating amounts of substrate (here peptides) and several concentrations of inhibitor. A double reciprocal plot shows a characteristic pattern for competitive inhibition. This experiment should be done.

For clarity, we have listed below the extensive amount of evidence that is presented in the manuscript that supports allosteric inhibition. In addition, we are also happy to present further evidence that is requested by the reviewer. As suggested, we performed a substrate saturation curve on the WT 20S proteasome using the substrate nLPnLD-amc in the absence and presence of the inhibitory oligomers. Instead of using double reciprocal plots, we used non-linear regression and the Michaelis-Menten equation to more accurately analyze the K_D and V_{max} of the two curves we produced. We found that the oligomers caused a decrease in the V_{max} and a small increase in the K_m (Supplementary Figure 8). Such inhibitors that decrease the V_{max} and increase the K_m are “mixed” allosteric inhibitors. This mode of inhibition is consistent with and further bolsters the large amount of data in the manuscript that support an allosteric mode of inhibition. Furthermore, the increasing K_m in this experiment is consistent with a mechanism that supports gate closure, which prevents substrates from reaching the active sites, thus decreasing their apparent affinity. We thank the reviewer for this recommendation and we have added these results to the manuscript: **Supp. Figure 8 (corresponds to Figure 3).**

Corresponding text added to the manuscript: *“To further confirm an allosteric mechanism of proteasome impairment, we performed a substrate saturation curve on the WT 20S proteasome with and without $A\beta^{*56}$ oligomers. We used non-linear regression and the Michaelis-Menten equation to analyze the K_D and V_{max} of the two curves. We found that the $A\beta^{*56}$ oligomers caused a decrease in the V_{max} and a slight increase in the K_m (Fig. S8), which is consistent with allosteric inhibition (i.e. mixed inhibition—a form of noncompetitive inhibition).”*

Data now presented in the manuscript that supports an allosteric mode of inhibition:

- 1) Oligomers are found to have an allosteric mechanism of inhibition, both decreasing the V_{max} and increasing the K_m of a peptide substrate.
- 2) 56kDa oligomers are far too large to enter the 13Å 20S translocation channel and thus cannot physically compete with the peptidase sites, which is required for text-book competitive inhibition.
- 3) PA26 and PA28 regulators that allosterically open the 20S gate via a non-HbYX dependent mechanism also blocks the inhibitory effects of oligomers.
- 4) Oligomers cannot inhibit proteasomes that do not have a functional gate (i.e. $\alpha3\Delta N$ 20S) and gating is an allosteric process.
- 5) Oligomers do not inhibit the proteasome as well when the gate is allosterically opened by HbYX containing peptides.
- 6) Oligomers cannot inhibit the 26S proteasome in the presence of ATP γ S, which more strongly stimulates allosteric gate-opening than does ATP.

The second technical question is related to the adequacy of the evidence for gate opening. Peptidase activity is commonly and conveniently used, but this surrogate for structure does not offer definitive evidence. That requires structural data. For 20S, negative stain EM is likely to suffice, but cryoEM would provide the higher resolution data that might be required to show partial or intermittent effects of inhibitor plus peptide on gate status. EM would suffice to answer the question for 20S.

We do not know of a negative stain method or publication that could accurately determine the state of the gate in the intact eukaryotic 20S proteasome. The only published finding showing a change in the state of the gate using negative stain EM that we are aware of is using isolated homoheptameric α -rings from the archaeal 20S proteasome. In these experiments, stain would collect inside alpha rings when all 7 N-termini had been deleted compared to WT α -rings. However, similar experiments have not been published for the eukaryotic 20S α -rings as they are heteroheptamers, and it is not trivial to isolate the heteroheptameric $\alpha1-7$ -ring. In addition, the state of the gate in isolated α -rings is not mechanistically relevant here since we are studying the mechanism of the intact 20S particle.

The claim that oligomers alter the ability of 26S to open the gate is more technically demanding, but also well within the capacity of cryoEM.

The reviewers point is well taken and the structure of the gate is well within the capacity of cryo-EM. In fact, many cryo-EM studies of the 26S proteasome have been published in the past five years. The issue is that all of these studies (+/-ATP, and +/- bound substrate) have found the 26S proteasome gate to be in the closed conformation. This was a surprise to some in the field. There are perhaps many reasons or explanations for this but sufficient to say gating dynamic in the 20S and 26S proteasome are not well understood at this time. Nevertheless, if cryo-EM always shows a closed gated proteasome, then addition of oligomers will not show a change in the state of the 20S gate meaning that these experiments would

not be useful. Determining the structure of the 20S-oligomer complex would be fantastic indeed, but this would obviously require a separate independent study, and it is well outside the scope of this already very mature and lengthy biochemical study.

We would also like to note that one very recent cryo-EM study (Wehmer et al., 2017, PNAS) of the 26S proteasome bound to a ATP ground state analog, ATP/ADP-BeFx, has shown an apparent open-state of the 20S channel in 9% of the particles they imaged (“s4 state”). However, the ATP/ADP-BeFx analog mimics the slowly hydrolysable analog ATP γ S, which we have also used in our manuscript. We found that oligomers could not inhibit the 26S proteasome when in the presence of ATP γ S (Fig 6A). This is consistent with our mechanism of action that we have described. Therefore, although this method does generate a small portion of particles with an open gate, adding the oligomers to monitor gate closing would not be successful since oligomers do not inhibit this ATP γ S state of the proteasome.

These cryo-EM studies demonstrate that the 20S gate is not either open or closed (as very active complexes appear closed via cryo-EM), but is instead highly dynamic. In fact, Luis Kay’s studies of the archaeal proteasome have shown that the 20S gate fluctuates between open and closed on the milli-second time scale. Because Cryo-EM is an averaging technique, it cannot efficiently report on the changes to the dynamics of opening and closing of the gate. Observing and quantifying these opening/closing dynamics is necessary to observe a change in the proportion of open and closed states. Therefore, to date, the only methods available to us that directly assay the dynamics of the closed/open state of the gate in the eukaryotic proteasome (i.e. to allow observation of open/closed state changes) are methods that monitor the entry rates of substrates, which has been historically useful and accurate based on many prior published studies. In this study, we have employed this highly useful method to directly assay gating dynamics of the proteasome.

Some minor points-

Related to fig. 1. Do A11+ oligomers maintain oligomeric structure with antibody? Do they remain soluble with antibody?

If oligomers were to lose their oligomeric structure when bound by the antibody, then the antibody could not function as an oligomer specific antibody. In addition, dot blots retain a stable signal after incubation with the A11 antibody. Based on these points we are confident that oligomers retain their structure when bound by the antibody.

To determine if the A11+ oligomers remain soluble with the A11 antibody, we performed an aggregation assay using dynamic light scattering (DLS). We found that the intensity of light scattering of the oligomer-A11 antibody complex was approximately equal to the sum of the light scattering intensity of the oligomer by itself and the A11 antibody by itself (see adjacent figure). This result indicates that no aggregation takes place due to oligomer-antibody binding. We used a similar amount of heat denatured aggregated albumin (HD-BSA) as a positive control for aggregation. This aggregated albumin had a signal that was approximately 16 times larger than the signal from A11 + oligomer. In addition, there were no visible signs of precipitation when the A11 antibody was incubated with the oligomers. As a side note, we also know that the oligomers retain their structure and retain solubility when bound by the N-terminal A β antibody since these oligomer/antibody complex still inhibits the proteasome.

Various figures. Absolute values of 100% peptidase activity (no inhibitor) are shown for WT and gate mutant 20S. What are the absolute values on some common scale of activity? We have addressed this point in detail above.

Data described as in sup. Fig. 7A are actually in figure 8.
Correction made.

For the section headed "Toxic Oligomers impair 26S activity in brain lysates", the data referred to in this section seem to be missing in the MS provided for review.

This data was presented in Figure 5 (E-H). We apologize for not inserting the Figure references in the manuscript text under the "Toxic Oligomers impair 26S activity in brain lysates" section header, which was likely confusing. As described above, we altered the order in which these data are presented in the manuscript text and figures, and we now believe that has improved the flow and readability of the manuscript. We believe that this addresses the reviewer's concern.

Reviewers' comments:

Reviewer #1 (Remarks to the Author):

The authors have addressed my technical concerns and corrected the mistakes in the manuscripts but my concern when it comes to the technical advance remains. The mechanism that is revealed in this study is very similar to the one shown in an earlier study entitled “Misfolded PrP impairs the UPS by interaction with the 20S proteasome and inhibition of substrate entry” published in EMBO Journal in 2011. Despite the similarities and despite the fact that one of the corresponding author is shared first-author of the earlier study, the authors did not comment on this work in the first version of the manuscript, which I felt was incorrect as it appears to be highly relevant for their present work. The authors argue now that their study has the desired conceptual advance for Nature Communications because 1) it shows that the same mechanism also applies for other neurodegeneration proteins (which are more common than prion pathology) and 2) they performed a more thorough study (using purer oligomers and human instead of yeast proteasomes). I am unconvinced by these arguments. The fact that the study comes to a similar conclusion with more pure reagents can hardly be an argument. On the other hand, it is appreciated that this study broadens the relevance of these findings to other, more common neurodegenerative diseases but mechanistically that does not make it very revealing. It also leaves me puzzled why they did not discuss this study in their original submission and raises to me the suspicion that they did so to boost the novelty of their present work. In the revision, they refer to the paper in their discussion but given its relevance the only appropriate way to present it is in the introduction where they should mention that an earlier study showed that prion aggregates prevent the opening of the proteasome and thereby impairs degradation. I realize that it will be a bit of a spoiler for their new data but relevant background should be presented in the introduction and not in the discussion. Altogether I have no technical concerns but remain convinced that the technical advance is not sufficient for Nature Communications. Moreover, I am also still concerned with the way the earlier study is presented in the revision.

Reviewer #3 (Remarks to the Author):

According to my appreciation, the letter of rebuttal adequately responds to the concerns of the three reviewers.

Some minor residual points.

Reviewer 2

Response to-

5. The discussion may benefit by some consideration about the physiological relevance of the concentrations of a particular oligomer used to inhibit the proteasome?

This comment seems to me to have been misunderstood by the authors. The authors argue correctly that oligomers have high affinity, and therefore will be largely bound to proteasomes regardless of oligomer concentration. But the comment of reviewer 2 regarding physiology is more plausibly directed to the question of what fraction of the proteasome population is undergoing inhibition. This is of course dependent on whether oligomers of interest approach proteasome concentration, here postulated as micromolar. As the proteasome activity is regarded as present in great excess of biological need under basal cellular conditions, the question is very relevant to physiologic relevance of the inhibition described.

Reviewer 3

Response to-

2. The native activity of the proteasome is as a protease....

Relevant data is fig. 5 G&H, as casein (Fig. E&F) is weakly structured and both a 20S or 26S substrate. Because the experiment was done in crude brain extract containing multiple proteases, controls should include no ATP and an inhibitor more specific than MG132.

Response to question of whether inhibition is allosteric.

Kinetic data now shown (Supl. Fig. 8) is consistent with allostery. It is not clear why it is said that "... oligomers caused a decrease in the V_{max} and a small increase in the K_m (Supplementary Figure 8)." Changes in both kinetic parameter are about two-fold.

Allostery is a term used a bit loosely. Formally it requires demonstrating that the regulator acts at a site distinct from the substrate association site(s). This is too high a standard to apply here, so let's give this question a pass.

Philip Coffino

Reviewer #1 (Remarks to the Author):

The authors have addressed my technical concerns and corrected the mistakes in the manuscripts but my concern when it comes to the technical advance remains. The mechanism that is revealed in this study is very similar to the one shown in an earlier study entitled "Misfolded PrP impairs the UPS by interaction with the 20S proteasome and inhibition of substrate entry" published in EMBO Journal in 2011. Despite the similarities and despite the fact that one of the corresponding author is shared first-author of the earlier study, the authors did not comment on this work in the first version of the manuscript, which I felt was incorrect as it appears to be highly relevant for their present work. The authors argue now that their study has the desired conceptual advance for Nature Communications because 1) it shows that the same mechanism also applies for other neurodegeneration proteins (which are more common than prion pathology) and 2) they performed a more thorough study (using purer oligomers and human instead of yeast proteasomes). I am unconvinced by these arguments. The fact that the study comes to a similar conclusion with more pure reagents can hardly be an argument. On the other hand, it is appreciated that this study broadens the relevance of these findings to other, more common neurodegenerative diseases but mechanistically that does not make it very revealing. It also leaves me puzzled why they did not discuss this study in their original submission and raises to me the suspicion that they did so to boost the novelty of their present work. In the revision, they refer to the paper in their discussion but given its relevance the only appropriate way to present it is in the introduction where they should mention that an earlier study showed that prion aggregates prevent the opening of the proteasome and thereby impairs degradation. I realize that it will be a bit of a spoiler for their new data but relevant background should be presented in the introduction and not in the discussion. Altogether I have no technical concerns but remain convinced that the technical advance is not sufficient for Nature Communications. Moreover, I am also still concerned with the way the earlier study is presented in the revision.

We would like to point out that, though the details were not discussed, the Deriziotis et. al. study was discussed and referenced in the introduction of all version of our manuscript as we indeed thought it was important to reference (see quote)
Introduction, 3rd paragraph:

"Several groups have provided evidence that aggregated proteins from neurodegenerative diseases interact with and impair proteasome function²⁷⁻³⁷"

Secondly, it appears that the reviewer may have missed that in our revised manuscript (based on his/her comments) we discussed the Deriziotis et. al. study in more detail in the first paragraph of the Results section, were we thought it logically fit best, prior to any discussion of our own experiments.

To fully address the reviewer's further concerns, we have now moved the description of the Prion paper in the results to the introduction section per the reviewer's request. In the third paragraph of the introduction we begin to introduce the reader to the concept of protein aggregates impairing the proteasome. We now follow this with a now more detailed description of the Deriziotis et. al. study. The text now reads:

"Several groups have provided evidence that aggregated proteins from neurodegenerative diseases interact with and impair proteasome function²⁷⁻³⁷. However,

Authors' response to reviewers' comments

it is not clear what specific types of aggregates impair the proteasome, and a mechanistic understanding of how they do so has not been elucidated. Though, one study has been able to show that heterogeneous aggregates of the mouse prion protein, PrP^{Sc}, reduced substrate entry by decreasing proteasomal gating³⁶."

Next, we present the knowledge gap that we are aiming to address in our study:

"Despite these many efforts, an understanding of why and how the proteasome is so generally impaired in neurodegenerative disease has remained elusive".

Here we are pointing out our broad goal to understand why the proteasome is impaired across a wide spectrum of neurodegenerative diseases. We feel this explicitly and appropriately credits the prior work, as well as lays the foundation for our experimental approach.

Reviewer #3 (Remarks to the Author):

According to my appreciation, the letter of rebuttal adequately responds to the concerns of the three reviewers.

Some minor residual points.

Reviewer 2

Response to-

5. The discussion may benefit by some consideration about the physiological relevance of the concentrations of a particular oligomer used to inhibit the proteasome?

This comment seems to me to have been misunderstood by the authors. The authors argue correctly that oligomers have high affinity, and therefore will be largely bound to proteasomes regardless of oligomer concentration. But the comment of reviewer 2 regarding physiology is more plausibly directed to the question of what fraction of the proteasome population is undergoing inhibition. This is of course dependent on whether oligomers of interest approach proteasome concentration, here postulated as micromolar. As the proteasome activity is regarded as present in great excess of biological need under basal cellular conditions, the question is very relevant to physiologic relevance of the inhibition described.

To address this concern raised by Reviewer 3 (we believe), regarding reviewer 2's comments, we have added the below text (in blue italics) following our discussion of the intracellular concentration of proteasomes and A β :

"While the physiological concentration of A11+ oligomers in neurons is unknown, if we consider that the affinity constant for the oligomers is low nM, and that the cellular concentration of the 20S is estimated to be low micromolar⁷⁰ then, with respect to this binding reaction, the 20S is saturating in the cell. This implies toxic oligomers will bind to the 20S irrespective of their cellular concentration. Which begs the question: are physiological levels of A11+ oligomers sufficient to impact protein degradation? Using laser capture microdissection and isolation of hippocampal pyramidal neurons from sporadic Alzheimer's Disease cases, Hashimoto et al. (2010) determined the intraneuronal concentration of A β 42 to be 3 μ M⁷¹, but what proportion of the intracellular A β 42 is in oligomeric form is not known. Furthermore, Kisselev et. al. (2006) showed that the amount of proteasome inhibitor,

Velcade™ that is used to treat multiply myeloma only inhibits protein degradation by about 10-25%⁷². This result demonstrates that a relatively small alteration of protein breakdown can have a substantial impact on cell death. Consistent with this reasoning, stereotaxic unilateral infusion of lactacystin (a selective proteasome inhibitor) into the substantia nigra pars compacta of rats caused neurodegenerative disease like symptoms²². However, the percentage of proteasomes that must be active in neurons to maintain normal proteostasis is not known and thus we could only speculate about what level of intracellular A11 oligomers would be required to impact neuronal function. Nevertheless, as protein degradation begins to suffer as oligomers accumulate, the level of proteasome impairment is expected to increase exponentially as more proteins accumulate and oligomerize. Such a model would be expected to exhibit exponential progression kinetics, which coincides with the exponential deterioration that is observed over decades in most neurodegenerative diseases. These results build confidence that such oligomers in neurons could impair proteasome function enough to contribute to the progression of these neurodegenerative diseases.

This short discussion above was previously included in the result section. Addressing the reviewers concerns required considerably more elaboration so we felt that with the new additions that it was better suited for the discussion, so we have now moved this paragraph there.

Reviewer 3

Response to-

2. The native activity of the proteasome is as a protease....

Relevant data is fig. 5 G&H, as casein (Fig. E&F) is weakly structured and both a 20S or 26S substrate. Because the experiment was done in crude brain extract containing multiple proteases, controls should include no ATP and an inhibitor more specific than MG132.

As the reviewer points out, casein is a substrate of the both the 20S and 26S. Both 20S and 26S forms of the proteasome are present in brain tissue and important for neuronal function. If we performed the experiment in brain lysates without ATP, the 20S proteasomes would still have capacity to degrade casein and the oligomers still have the capacity to inhibit the 20S (as we have shown). Based on this reasoning, an identical experiment without ATP would not provide useful results.

With regards to the specificity of MG132, thousands of published studies have used MG132 as a specific proteasome inhibitor for *in vitro* studies. However, no proteasome inhibitor lacks off-target effects and as the reviewer correctly pointed out MG132, isn't specific to only the proteasome. MG132 can also inhibit calpains (at the concentrations that we used), which are indeed capable of degrading casein. We considered this fact during our experimental design, and chose to address this issue in a different, and we believe better, way. We elected to repeat the brain lysate experiments with a folded and ubiquitinated substrate that is specific to the 26S proteasome (Ub₄(lin)GFP-35). Calpains do not have unfoldase activity and therefore are not expected to be able degrade Ub₄(lin)GFP-35. Our experiments with this GFP substrate also showed that our oligomers could inhibit its degradation in brain lysates (Figure 5). We believe this is the most stringent approach to demonstrate 26S proteasome activity. We believe that our demonstration that these oligomers inhibit purified 20S and 26S proteasomes and that they inhibit two different protein substrates in brain lysates (one of which is ubiquitin and unfolding dependent), both of which completely responded to inhibition by the classical proteasome

Authors' response to reviewers' comments

inhibitor MG132, conclusively demonstrates that our oligomers can inhibit proteasome function in brain lysates. We believe this clarification satisfies the reviewers concern.

Response to question of whether inhibition is allosteric.

Kinetic data now shown (Supl. Fig. 8) is consistent with allostery. It is not clear why it is said that "... oligomers caused a decrease in the Vmax and a small increase in the Km (Supplementary Figure 8)." Changes in both kinetic parameter are about two-fold. Allostery is a term used a bit loosely. Formally it requires demonstrating that the regulator acts at a site distinct from the substrate association site(s). This is too high a standard to apply here, so let's give this question a pass.

To satisfy the reviewers concern we have removed the word "*small*" from this sentence.